# Information-theoretic Limits of Online Classification with Noisy Labels

**Changlong Wu    Ananth Grama    Wojciech Szpankowski**
CSoI, Purdue University
wuchangl@hawaii.edu, {ayg,szpan}@purdue.edu

## Abstract

We study online classification with general hypothesis classes where the true labels are determined by some function within the class, but are corrupted by *unknown* stochastic noise, and the features are generated adversarially. Predictions are made using observed *noisy* labels and noiseless features, while the performance is measured via minimax risk when comparing against *true* labels. The noisy mechanism is modeled via a general noisy *kernel* that specifies, for any individual data point, a set of distributions from which the actual noisy label distribution is chosen. We show that minimax risk is tightly characterized (up to a logarithmic factor of the hypothesis class size) by the *Hellinger gap* of the noisy label distributions induced by the kernel, *independent* of other properties such as the means and variances of the noise. Our main technique is based on a novel reduction to an online comparison scheme of two-hypotheses, along with a new *conditional* version of Le Cam-Birgé testing suitable for online settings. Our work provides the first comprehensive characterization for noisy online classification with guarantees that apply to the ground truth while addressing *general* noisy observations.

## 1   Introduction

Learning from noisy data is a fundamental problem in many machine learning applications. Noise can originate from various sources, including low-precision measurements of physical quantities, communication errors, or noise intentionally injected by methods such as differential privacy. In such cases, one typically learns by training on *noisy* (or observed) data while aiming to build a model that performs well on the *true* (or latent) data. This paper focuses on *online learning* [20] from noisy labels, where one receives noiseless, *adversarially* generated features and corresponding noisy labels sequentially, and predicts the *true* labels as the data arrive.

Online learning has been primarily studied in the *agnostic* setting [1, 19, 7], where one receives the labels in their plain (noise-free) form and the prediction risk is evaluated on the *observed* labels. It is typically assumed that both the features and observed labels are generated adversarially, and prediction quality is measured via the notion of *regret*, which compares the actual cumulative risk incurred by the predictor with the minimal cumulative risk incurred by the best expert in a hypothesis class. While this approach is mathematically appealing, it does not adequately characterize online learning scenarios when our goal is to achieve good performance with respect to *grand truth* data that may be different from the observed (noisy) ones.

This paper considers an online learning scenario that differs from classical *agnostic* online learning in two aspects: (i) we assume that the noisy labels are derived from a (semi-) *stochastic* mechanism rather than from pure adversarial selections; (ii) our prediction risk is evaluated on the *true* labels, not *noisy* observations. To better motivate the study of such a scenario, we consider the following example first introduced by Ben-David et al. [1]:

38th Conference on Neural Information Processing Systems (NeurIPS 2024).

**Example 1.** *Let $\mathcal{H} \subset \{0,1\}^{\mathcal{X}}$ be a finite hypothesis class. Consider the following online learning game between Nature/Adversary and Learner that is played over a time horizon $T$. At the start, Nature fixes a ground truth classifier $h \in \mathcal{H}$. At each time step $t \leq T$, Nature adversarially selects feature $\mathbf{x}_t \in \mathcal{X}$ and reveals it to the learner. The learner makes a prediction $\hat{y}_t$ based on prior features $\mathbf{x}^t = \{\mathbf{x}_1, \cdots, \mathbf{x}_t\}$ and noisy labels $\tilde{y}^{t-1} = \{\tilde{y}_1, \cdots, \tilde{y}_{t-1}\}$. Nature then selects a (unknown) noise parameter $\eta_t \in [0, \eta]$ for some given $\eta$ (known to learner), and generates [1]:*

$$\tilde{y}_t = \mathsf{Bernoulli}(\eta_t) \oplus y_t,$$

*where $\oplus$ denotes binary addition and $y_t = h(\mathbf{x}_t)$ is the* true *label. It was shown in [1, Thm 15] that there exists a predictor $\hat{y}^T$ such that:*

$$\sup_{h \in \mathcal{H}, \mathbf{x}^T \in \mathcal{X}^T} \mathbb{E}\left[\sum_{t=1}^{T} 1\{\hat{y}_t \neq h(\mathbf{x}_t)\}\right] \leq \frac{\log|\mathcal{H}|}{1 - 2\sqrt{\eta(1-\eta)}}. \tag{1}$$

Note that the risk in (1) is significant, as the error introduced by noise to the true labels increases linearly as $\eta T$, yet the risk remains *independent* of the time horizon $T$. This mirrors the *fast rates* known in the PAC learning literature when *benign noise* is present. Despite its foundational nature, the understanding of this phenomenon beyond simple Massart's noise has been largely unexplored.

This paper introduces a novel online learning framework for modeling *general* noisy mechanisms. In particular, it encompasses (1) as a very specific instance and provides a clear and comprehensive characterization of the underlying paradigm. Formally, let $\mathcal{Y}$ be the set of true (latent) labels and $\tilde{\mathcal{Y}}$ be the set of noisy (observed) labels, which we assume are finite and of size $N, M$, respectively. Let $\mathcal{X}$ be the feature space. We model the noisy mechanism by a *noisy kernel*:

$$\mathcal{K} : \mathcal{X} \times \mathcal{Y} \to 2^{\mathcal{D}(\tilde{\mathcal{Y}})}, \tag{2}$$

where $\mathcal{D}(\tilde{\mathcal{Y}})$ is the set of all distributions over $\tilde{\mathcal{Y}}$. That is, the kernel $\mathcal{K}$ maps each pair $(\mathbf{x}, y)$ to a *subset* $\mathcal{Q}_y^{\mathbf{x}} := \mathcal{K}(\mathbf{x}, y) \subset \mathcal{D}(\tilde{\mathcal{Y}})$ of distributions over $\tilde{\mathcal{Y}}$. Observe that the noisy kernel provides a compact way of modeling *noisy label* distributions without explicitly referring to the *noise*. This is more convenient for our discussion, as ultimately the statistical information is solely determined by the noisy label distributions.

For any given $\mathcal{H} \subset \mathcal{Y}^{\mathcal{X}}$ and kernel $\mathcal{K}$, we consider the following *robust (noisy) online classification* scenario: Nature first selects $h \in \mathcal{H}$; at each time step $t$, Nature chooses (adversarially) $\mathbf{x}_t \in \mathcal{X}$ and reveals it to the learner; the learner then makes a prediction $\hat{y}_t$, based on the features $\mathbf{x}^t$ and *noisy* labels $\tilde{y}^{t-1}$; an *adversary* then selects a distribution $\tilde{p}_t \in \mathcal{Q}_{h(\mathbf{x}_t)}^{\mathbf{x}_t}$, samples $\tilde{y}_t \sim \tilde{p}_t$ and reveals $\tilde{y}_t$ to the learner. Let $\Phi$ and $\Psi$ be the strategies of the learner and Nature/adversary, respectively. The goal of the learner is to minimize the following expected minimax *risk*:

$$\tilde{r}_T(\mathcal{H}, \mathcal{K}) = \inf_{\Phi} \sup_{\Psi} \mathbb{E}\left[\sum_{t=1}^{T} 1\{h(\mathbf{x}_t) \neq \hat{y}_t\}\right], \tag{3}$$

where $\hat{y}_t = \Phi(\mathbf{x}^t, \tilde{y}^{t-1})$. Note that the adversarial selection of distribution $\tilde{p}_t$ from the kernel set $\mathcal{Q}_{h(\mathbf{x}_t)}^{\mathbf{x}_t}$ provides more flexibility for modeling scenarios when the noisy label distribution changes even with the same true label, such as the Massart's noise in Example 1. We refer to Section 2 for a more complete specification of our setting.

## 1.1 Main Contributions

Our main contributions in this paper establish the *fundamental limits* of minimax risk in (3) by providing nearly matching lower and upper bounds across a wide range of hypothesis classes $\mathcal{H}$ and noisy kernels $\mathcal{K}$. Observe that, to allow for non-trivial prediction rules, the induced noisy label distributions must be statistically *distinguishable* for distinct true labels. To formalize this intuition, we define, for any noisy kernel $\mathcal{K}$ and feature $\mathbf{x} \in \mathcal{X}$, the *Hellinger* gap as $\gamma_{\mathrm{H}}(\mathbf{x}) = \inf_{y \neq y' \in \mathcal{Y}} \inf_{p \in \mathcal{Q}_y^{\mathbf{x}}, q \in \mathcal{Q}_{y'}^{\mathbf{x}}} \{H^2(p, q)\}$, where $H^2(p, q) = \sum_{m=1}^{M}(\sqrt{p[m]} - \sqrt{q[m]})^2$ is the squared Hellinger distance. That is, $\gamma_{\mathrm{H}}(\mathbf{x})$ measures the minimal squared Hellinger distance of the induced noisy label distributions over all distinct true labels.

Our main result (see also Theorem 2) can be summarized as follows:

---

[1] This is also known as Massart's noise in the literature.

**Theorem 1.** *Let $\mathcal{H} \subset \mathcal{Y}^{\mathcal{X}}$ be any finite class, and $\mathcal{K}$ be any noisy kernel such that $\inf_{\mathbf{x} \in \mathcal{X}} \gamma_{\mathrm{H}}(\mathbf{x}) \geq \gamma_{\mathrm{H}}$ for some $\gamma_{\mathrm{H}} > 0$, and $\mathcal{Q}_y^{\mathbf{x}} \subset \mathcal{D}(\tilde{\mathcal{Y}})$ is closed and convex for all $\mathbf{x}, y$. Then:*

$$\tilde{r}_T(\mathcal{H}, \mathcal{K}) \leq O\left(\frac{\log^2 |\mathcal{H}|}{\gamma_{\mathrm{H}}}\right).$$

*Moreover, for any $K \in \mathbb{N}$ and any kernel $\mathcal{K}$ with at least $\log K$ features $\mathbf{x} \in \mathcal{X}$ for which $\gamma_{\mathrm{H}}(\mathbf{x}) \leq \gamma_{\mathrm{H}}$, there exists a class $\mathcal{H}$ of size $K$ that satisfies: $\tilde{r}_T(\mathcal{H}, \mathcal{K}) \geq \Omega\left(\frac{\log |\mathcal{H}|}{\gamma_{\mathrm{H}}}\right).$*

Theorem 1 shows that the *Hellinger* gap is the *right* characterization for the minimax risk upto at most a logarithmic factor. Moreover, the risk bound depends solely on the gap parameter $\gamma_{\mathrm{H}}$ and $\log |\mathcal{H}|$, *independent* of time horizon $T$, the size of $\mathcal{Y}$ and $\tilde{\mathcal{Y}}$, and the properties of noise such as means and variances. For the bounded Bernoulli noise in Example 1, the set $\mathcal{Q}_y^{\mathbf{x}}$ corresponds to Bernoulli distributions with parameters in $[0, \eta]$ if $y = 0$ and in $[1 - \eta, 1]$ if $y = 1$, leading to the Hellinger gap $\gamma_{\mathrm{H}} = 1 - 2\sqrt{\eta(1 - \eta)}$. This matches the dependency on $\eta$ in Example 1 [2]. However, our result holds for *any* noisy kernel. For instance, if we shift $\mathcal{Q}_0^{\mathbf{x}}$ to Bernoulli distribution with parameter 0 and $\mathcal{Q}_1^{\mathbf{x}}$ with parameters in $[1 - 2\eta, 1]$, then $\gamma_{\mathrm{H}} = 1 - \sqrt{2\eta} = \Theta(1 - 2\eta)$. This is tighter than the dependency on $\eta$ in Example 1 (for $\eta \to \frac{1}{2}$), since $1 - 2\sqrt{\eta(1 - \eta)} = \Theta((1 - 2\eta)^2)$.

Our main proof technique for establishing Theorem 1 is based on a novel (black box) reduction to an online comparison scheme of two-hypotheses in $\mathcal{H}$, as demonstrated in Theorem 3. This allows us to reduce the noisy online *classification* problem to a *hypothesis testing* problem, which effectively decouples the *adversarial* property of the features from the *stochastic* property of the noisy labels. However, due to the adversarial selection of the noisy label distributions, the classical hypothesis testing techniques does not apply. To resolve this issue, we establish in Theorem 4, a generalization of the Le Cam-Birgé Test with *varying* conditional marginals for handling pairwise testing via the *Hellinger* gap, which is a result of independent interest.

**Tight dependency on $\log |\mathcal{H}|$.** Although the lower and upper bounds in Theorem 1 differ by a $\log |\mathcal{H}|$ factor, this is compensated by the fact that we are dealing with the most general classes and kernels. This can be tightened for various special cases. Indeed, for a class $\mathcal{H}$ with *binary* true labels and arbitrary noisy labels, we demonstrate in Theorem 5 that the minimax risk is upper bounded by $\frac{16 \log |\mathcal{H}|}{\gamma_{\mathrm{L}}}$, where $\gamma_{\mathrm{L}}$ is the $L^2$-gap that substitutes the Hellinger distance with $L^2$-distance in Theorem 1. This is proved via a novel reduction to online conditional distribution estimation under $L^2$-distance. Moreover, we demonstrate in Appendix G (Theorem 6) that the (optimal) $O(\frac{\log |\mathcal{H}|}{\gamma_{\mathrm{H}}})$ upper bound holds if $|\mathcal{Q}_y^{\mathbf{x}}| = 1$ for all $\mathbf{x}, y$, i.e., the noisy label distribution is *determined* by data.

## 1.2 Related Work

Online learning with noisy data was discussed in [6], which specifically considers generalized *linear* functions with zero-mean and bounded variance noises. Our work differs in that we focus on classification instead of regression. Moreover, our noisy model does not require that the noise be zero-mean. To the best of our knowledge, [1] is the only work that has specifically considered the classification task, but this was limited to bounded Bernoulli noise. From a technical standpoint, analogous ideas of pairwise comparison have been considered in differential privacy literature, such as in [11], but only in *batch* settings. The reduction to online conditional probability estimation was also explored in [10] within the context of *online decision making*. However, a distinguishing feature of our work is that our conditional probability estimation problem is necessarily *misspecified*, as our noisy label distributions are selected *adversarially* and are unknown a priori to the learner. Our problem setup is further related to *differentially private* conditional distribution learning, as in [26], and *robust hypothesis testing*, discussed in [17, Chapter 16]. Online conditional probability estimation has been widely studied, see [18, 3, 2, 4, 25, 24]. Conditional density estimation in the *batch* setting has also been extensively studied, see [12] for KL-divergence with misspecification and [9] for $L^2$ loss. Learning from noisy labels in the *batch* case was discussed in [16] (see also the references therein) by leveraging suitably defined proxy losses. There has been a long line of research on online prediction with *adversarial* observable labels in the *agnostic* formulation, see [5, 1, 19, 7].

---

[2]To the best of our knowledge, this Hellinger interpretation is not known in literature; the proof in [1] is based on *induction* without explaining on how the factor $1 - 2\sqrt{\eta(1 - \eta)}$ is obtained.

## 2 Notation and Preliminaries

Let $\mathcal{X}$ be a set of features (or instances), $\mathcal{Y}$ be a set of labels, and $\tilde{\mathcal{Y}}$ be a set of noisy observations. We assume throughout the paper that $|\mathcal{Y}| = N$ and $|\tilde{\mathcal{Y}}| = M$ for some integers $N, M \geq 2$. We denote $\mathcal{D}(\tilde{\mathcal{Y}})$ as the set of all probability distributions over $\tilde{\mathcal{Y}}$.

Let $\mathcal{H} \subset \mathcal{Y}^{\mathcal{X}}$ be a hypotheses class and $\mathcal{K}$ be a noisy kernel in (2). We consider the following robust online classification scenario: (1) *Nature* first selects some $h \in \mathcal{H}$; (2) At time $t$, Nature adversarially selects $\mathbf{x}_t \in \mathcal{X}$; (3) Learner predicts $\hat{y}_t \in \mathcal{Y}$, based on (noisy) history observed thus far (i.e., $\mathbf{x}^t, \tilde{y}^{t-1}$); (4) An *adversary* then selects $\tilde{p}_t \in \mathcal{Q}_{h(\mathbf{x}_t)}^{\mathbf{x}_t}$, and generates a *noisy* sample $\tilde{y}_t \sim \tilde{p}_t$.

The goal of the learner is to minimize the *cumulative error*: $\sum_{t=1}^{T} 1\{h(\mathbf{x}_t) \neq \hat{y}_t\}$.

Note that the cumulative error is a *random variable* that depends on all the randomness associated with the game. To remove the dependency on such randomness and to assess the fundamental limits of the prediction quality, we consider the following two measures [3]:

**Definition 1.** *Let $\mathcal{H} \subset \mathcal{Y}^{\mathcal{X}}$ be a set of hypotheses and $\mathcal{K} : \mathcal{X} \times \mathcal{Y} \to 2^{\mathcal{D}(\tilde{\mathcal{Y}})}$ be a noisy kernel. We denote by $\Phi$ the (possibly randomized) strategies of the* learner. *The* expected minimax risk *is:*

$$\tilde{r}_T(\mathcal{H}, \mathcal{K}) = \inf_{\Phi} \sup_{h \in \mathcal{H}, \mathbf{x}^T \in \mathcal{X}^T} \mathbb{Q}_{\mathcal{K}}^T \mathbb{E}_{\hat{y}^T} \left[ \sum_{t=1}^{T} 1\{h(\mathbf{x}_t) \neq \hat{y}_t\} \right], \tag{4}$$

*where* $\mathbb{Q}_{\mathcal{K}}^T \equiv \sup_{\tilde{p}_1 \in \mathcal{Q}_{h(\mathbf{x}_1)}^{\mathbf{x}_1}} \mathbb{E}_{\tilde{y}_1 \sim \tilde{p}_1} \cdots \sup_{\tilde{p}_T \in \mathcal{Q}_{h(\mathbf{x}_T)}^{\mathbf{x}_T}} \mathbb{E}_{\tilde{y}_T \sim \tilde{p}_T}$, *and* $\hat{y}_t \sim \Phi(\mathbf{x}^t, \tilde{y}^{t-1})$.

By *skolemization* [19], the operator $\mathbb{Q}_{\mathcal{K}}^T$ is equivalent to $\sup_{\tilde{p}} \mathbb{E}_{\tilde{y}^T \sim \tilde{p}}$, where $\tilde{p}$ runs over all (joint) distributions over $\tilde{\mathcal{Y}}^T$ such that $\forall t \in [T], \forall \tilde{y}^{t-1} \in \tilde{\mathcal{Y}}^{t-1}$ the *conditional* marginal $\tilde{p}_{\tilde{y}_t | \tilde{y}^{t-1}} \in \mathcal{Q}_{h(\mathbf{x}_t)}^{\mathbf{x}_t}$.

**Definition 2.** *Let $\mathcal{H}$, $\mathcal{K}$, and $\Phi$ be as in Definition 1. For any $\delta > 0$, the* high probability minimax risk *at confidence $\delta$ is the minimum quantity $B^{\delta}(\mathcal{H}, \mathcal{K}) \geq 0$ such that there exists a predictor $\Phi$ satisfying:*

$$\sup_{h \in \mathcal{H}, \mathbf{x}^T \in \mathcal{X}^T, \tilde{p}} \Pr_{\tilde{y}^T \sim \tilde{p}, \hat{y}^T} \left[ \sum_{t=1}^{T} 1\{h(\mathbf{x}_t) \neq \hat{y}_t\} \geq B^{\delta}(\mathcal{H}, \mathcal{K}) \right] \leq \delta, \tag{5}$$

*where $\tilde{p}$ is selected as in the discussion above and $\hat{y}_t \sim \Phi(\mathbf{x}^t, \tilde{y}^{t-1})$.*

Note that the kernel map $\mathcal{K}$ is generally *known* to the learner when constructing the predictor $\Phi$. However, the induced kernel sets $\mathcal{Q}_{h(\mathbf{x}_t)}^{\mathbf{x}_t}$ are not, since they depend on the *unknown* ground truth classifier $h$ and *adversarially* generated features $\mathbf{x}^T$. In certain cases, such as Theorem 3 and Example 4, the kernel map $\mathcal{K}$ is also *not* required to be known.

For any $\mathbf{x} \in \mathcal{X}$ and $y \in \mathcal{Y}$, we denote by $\mathcal{Q}_y^{\mathbf{x}}$ the set induced by a kernel. We can assume, w.l.o.g., that the $\mathcal{Q}_y^{\mathbf{x}}$s are *convex* and *closed* sets, since the adversary can select an arbitrary distribution from $\mathcal{Q}_y^{\mathbf{x}}$s at each time step, including randomized strategies that effectively sample from a mixture (i.e., convex combination) of distributions in $\mathcal{Q}_y^{\mathbf{x}}$s.

One must introduce some constraints on the kernel $\mathcal{K}$ in order to obtain meaningful results. To do so, we introduce the following *well-separation* condition:

**Definition 3.** *Let $L : \mathcal{D}(\tilde{\mathcal{Y}})^2 \to \mathbb{R}^{\geq 0}$ be any divergence, we say a kernel $\mathcal{K}$ is* well-separated *w.r.t. $L$ at scale $\gamma > 0$, if $\forall \mathbf{x} \in \mathcal{X}, \forall y, y' \in \mathcal{Y}$ with $y \neq y'$ we have $L(\mathcal{Q}_y^{\mathbf{x}}, \mathcal{Q}_{y'}^{\mathbf{x}}) \overset{\text{def}}{=} \inf_{p \in \mathcal{Q}_y^{\mathbf{x}}, q \in \mathcal{Q}_{y'}^{\mathbf{x}}} L(p, q) \geq \gamma$.*

**Example 2.** *For any $y \in \mathcal{Y}$, we specify a* canonical *distribution $p_y \in \mathcal{D}(\tilde{\mathcal{Y}})$. A natural noisy kernel would be to define $\mathcal{Q}_y^{\mathbf{x}} = \{p \in \mathcal{D}(\tilde{\mathcal{Y}}) : \mathsf{TV}(p, p_y) \leq \epsilon\}$, where $\mathsf{TV}$ denotes total variation. In this case, the kernel is well-separated with the gap $\gamma$ under total variation if $\min_{y \neq y' \in \mathcal{Y}} \mathsf{TV}(p_y, p_{y'}) \geq \gamma + 2\epsilon$. In particular, this subsumes Example 1 if, for $y \in \{0, 1\}$, we define $p_y$ as the distribution that assigns probability 1 to $y$, and take $\epsilon = \eta$, where the TV-gap equals $\gamma = 1 - 2\eta$.*

---

[3]We assume here the selection of $\tilde{p}^T$ and $\mathbf{x}^T$ are oblivious to the learner's action for simplicity. This is equivalent to the adaptive case if the learner's internal randomness are independent among different time steps by a standard argument from [5, Lemma 4.1], see also Appendix H.

## 3  Main Results

We begin by stating our main result of this paper.

**Theorem 2.** *Let $\mathcal{H} \subset \mathcal{Y}^{\mathcal{X}}$ be a finite class of size $K$, and $\mathcal{K}$ be a kernel that is well-separated at scale $\gamma_{\mathrm{H}}$ w.r.t. squared Hellinger divergence (Definition 3). Then, the high probability minimax risk (Definition 2) with confidence $\delta > 0$ is upper bounded by:*

$$B^{\delta}(\mathcal{H}, \mathcal{K}) \leq \frac{8 \log(4K/\delta) \log K}{\gamma_{\mathrm{H}}} + \log(2/\delta). \tag{6}$$

*Moreover, for any kernel $\mathcal{K}$ such that there exist at least $L$ distinct features $\mathbf{x} \in \mathcal{X}$ [4] for which $\inf_{y \neq y' \in \mathcal{Y}} H^2(\mathcal{Q}_y^{\mathbf{x}}, \mathcal{Q}_{y'}^{\mathbf{x}}) \leq \gamma_{\mathrm{H}}$, one can find a class $\mathcal{H}$ of size $K$ such that:*

$$\tilde{r}_T(\mathcal{H}, \mathcal{K}) \geq \Omega\left(\frac{\min\{L, \log K\}}{\gamma_{\mathrm{H}}}\right).$$

Observe that, the upper bound holds with *high probability* and the risk is independent of the time horizon (i.e., the so-called fast rates known in the PAC-learning literature). Moreover, the bound is *independent* of the size of $\mathcal{Y}$ and $\tilde{\mathcal{Y}}$. A simple integration argument yields the *expected* risk upper bound $\tilde{r}(\mathcal{H}, \mathcal{K}) \leq O\left(\frac{\log^2 K}{\gamma_{\mathrm{H}}}\right)$, which matches the lower bound upto only a $\log K$ factor. This demonstrates that, the *Hellinger* gap of the induced noisy label distributions is the *right* characterization for the minimax risk. Moreover, the Hellinger distance can be transformed from other $f$-divergences (such total variation) without depending on the size of $\tilde{\mathcal{Y}}$ [17, Chapter 7.6].

**Example 3.** *Let $\mathcal{K}$ be the kernel in Example 2. Let $\lambda = \min_{y \neq y' \in \mathcal{Y}} \mathsf{TV}(p_y, p_{y'})$. Hence, the kernel is well-separated with TV-gap $\lambda - 2\epsilon$. Since $H^2(p, q) \geq \mathsf{TV}(p, q)^2$ [17, Eq. 7.22], the Hellinger gap is lower bounded by $(\lambda - 2\epsilon)^2$. Invoking Theorem 2, we have for* any *hypothesis class $\mathcal{H}$, the following risk upper bound holds: $B^{\delta}(\mathcal{H}, \mathcal{K}) \leq O\left(\frac{\log |\mathcal{H}| \log(|\mathcal{H}|/\delta)}{(\lambda - 2\epsilon)^2}\right)$.*

The rest of this section is devoted to establishing Theorem 2. Our main proof technique is based on a novel reduction to pairwise testing of two hypotheses as developed in Section 3.1, along with explicit testing rules in Section 3.2 based on a novel *conditional* version of Le Cam-Birgé testing.

### 3.1  Reduction to Pairwise Comparison: a Generic Approach

We first introduce the following key technical concept. Recall that our robust online classification problem is completely determined by the tuple $(\mathcal{H}, \mathcal{K})$.

**Definition 4.** *A problem $(\mathcal{H}, \mathcal{K})$ is said to be* pairwise testable *with confidence $\delta > 0$ and error bound $C(\delta) \geq 0$ if, for* any *pair $h_i, h_j \in \mathcal{H}$, the sub-problem $(\{h_i, h_j\}, \mathcal{K})$ admits a predictor (i.e., pairwise tester) $\Phi_{i,j}$ that achieves cumulative risk $\leq C(\delta)$ w.p. $\geq 1 - \delta$ (see Definition 2).*

Clearly, any prediction rule for $(\mathcal{H}, \mathcal{K})$ serves as a pairwise testing rule for all the sub-problems $(\{h_i, h_j\}, \mathcal{K})$ with $h_i, h_j \in \mathcal{H}$. Perhaps surprisingly, we will show in this section that any pairwise testing rules for the sub-problems can also be converted into a prediction rule for $(\mathcal{H}, \mathcal{K})$, incurring only an additional logarithmic factor on the risk bounds.

To this end, suppose that the tuple $(\mathcal{H}, \mathcal{K})$ is *pairwise testable* and the class $\mathcal{H} = \{h_1, \cdots, h_K\}$ is finite with size $K$. Let $\Phi_{i,j}$ be the testing rule (will be constructed in Section 3.2) for $h_i, h_j$ with error bound $C(\delta)$ and confidence $\delta > 0$. Let $\mathbf{x}^T, \tilde{y}^T$ be any realization of problem $(\mathcal{H}, \mathcal{K})$. We define, for any $h_i \in \mathcal{H}$ and $t \in [T]$, a *surrogate loss* vector:

$$\forall j \in [K], \ \mathbf{v}_t^i[j] = 1\{\Phi_{i,j}(\mathbf{x}^t, \tilde{y}^{t-1}) \neq h_i(\mathbf{x}_t)\}. \tag{7}$$

That is, the loss $\mathbf{v}_t^i[j] = 1$ if and only if the test $\Phi_{i,j}(\mathbf{x}^t, \tilde{y}^{t-1})$ differs from $h_i(\mathbf{x}_t)$. Given access to testers $\Phi_{i,j}$s, our prediction rule for $(\mathcal{H}, \mathcal{K})$ is then presented in Algorithm 1.

---

[4]This is a very mild assumption. For instance, if the kernel is independent of the features (such as Example 1), we have $L = |\mathcal{X}|$. The lower bound gives $\Omega(\frac{\log K}{\gamma_{\mathrm{H}}})$ as long as $|\mathcal{X}| \geq \log K$.

**Algorithm 1:** Predictor via Pairwise Hypothesis Testing
___
**Input**: Class $\mathcal{H} = \{h_1, \cdots, h_K\}$, pairwise testers $\Phi_{i,j}$ for $i, j \in [K]$ and error bound $C$
Set $S^1 = \{1, \cdots, K\}$;
**for** $t = 1, \cdots, T$ **do**
    Receive $\mathbf{x}_t$;
    Sampling index $\hat{k}_t$ from $S^t$ *uniformly* and make prediction: $\hat{y}_t = h_{\hat{k}_t}(\mathbf{x}_t)$;
    Receive noisy label $\tilde{y}_t$;
    Set $S^{t+1} = \emptyset$;
    **for** $i \in S^t$ **do**
        Compute $l_t^i = \max_{j \in [K]} \sum_{r=1}^t \mathbf{v}_r^i[j]$, where $\mathbf{v}_t^i[j]$ is defined in (7);
        **if** $l_t^i \leq C$ **then**
            Update $S^{t+1} = S^{t+1} \cup \{i\}$;
___

**Theorem 3.** *Let $\mathcal{H} \subset \mathcal{Y}^{\mathcal{X}}$ be any hypothesis class of size $K$ and $\mathcal{K}$ be any noisy kernel. If $(\mathcal{H}, \mathcal{K})$ is pairwise testable with error bound $C(\delta)$ as in Definition 4, then for any $\delta > 0$, the predictor in Algorithm 1 with $C = C(\delta/(2K))$ achieves the* high probability *minimax risk (Definition 2):*

$$B^\delta(\mathcal{H}, \mathcal{K}) \leq 2(1 + 2C(\delta/(2K)) \log K) + \log(2/\delta). \tag{8}$$

*Sketch of Proof.* At a high level, our goal is to identify the ground truth classifier $h_{k^*}$ using the testing results of $\Phi_{i,j}$s. Note that pairwise testability implies, w.p. $\geq 1 - \delta$, the errors made by tester $\Phi_{k,k^*}$ on $h_{k^*}$ is upper bounded by $C(\delta/2K)$ for all $k \in [K]$ simultaneously. However, for any other pair $i, j \neq k^*$, the tester $\Phi_{i,j}$ does not provide any guarantees, since the samples used to test $h_i, h_j$ originate from $h_{k^*}$ and is not *realizable* for $\Phi_{i,j}$. The key technical challenge is to extract the testing results for $\Phi_{k,k^*}$ from the other irrelevant tests (i.e., $\Phi_{i,j}$ with $k^* \notin \{i, j\}$), even when the $k^*$ is *unknown*. This is resolved by our definition of $l_t^i$ in Algorithm 1, which computes for each $i$ the *maximum* testing loss over all of its competitors. This ensures that, for the ground truth $k^*$, the loss $l_t^{k^*} \leq C(\delta/2K)$. While for any other $i \neq k^*$, we have $l_t^i \geq \sum_{r=1}^t \mathbf{v}_r^i[k^*] \geq \sum_{r=1}^t \mathbf{1}\{h_i(\mathbf{x}_r) \neq h_{k^*}(\mathbf{x}_r)\} - C(\delta/2K)$. Therefore, any hypothesis $h_i$ for which $l_t^i > C(\delta/2K)$ cannot be the ground truth. Algorithm 1 then maintains an index set $S^t$ that eliminates all $h_i$ for which $l_t^i > C(\delta/2K)$, and makes prediction $\hat{y}_t = h_{\hat{k}_t}(\mathbf{x}_t)$ with $\hat{k}_t$ sampling *uniformly* from $S^t$.

To derive the risk bound, we use a *potential*-based analysis that relates the size of $S^t$s with the prediction error $\mathbf{1}\{h_{k^*}(\mathbf{x}_t) \neq \hat{y}_t\}$. The intuition behind the analysis is that if $\mathbb{E}[\mathbf{1}\{h_{k^*}(\mathbf{x}_t) \neq \hat{y}_t\}]$ is large, then there will be many elements $i \in S^t$ for which $h_i(\mathbf{x}_t) \neq h_{k^*}(\mathbf{x}_t)$, and thus the loss $l_t^i$ will (potentially) increase. Since Algorithm 1 constructs $S^{t+1}$ by eliminating all $i \in S^t$ for which $l_t^i > C(\delta/2K)$, one can therefore bound the prediction error by the *change* in the size of $S^t$s. The key technical challenge here is to control the hypotheses that differ from $k^*$ but for which the tester $\Phi_{k,k^*}$ errs, which is resolved by carefully defining a *potential* function. The claimed upper bound then follows by a similar argument as [14, Thm 2]. See Appendix B for complete proof. □

Note that, the reduction of Theorem 3 is *general* and does not rely on specific properties of the kernel $\mathcal{K}$ (such as the well-separation condition). It provides a *black box* reduction that converts any pairwise testing rule for two-hypotheses to a general online classification rule that introduces only a logarithmic factor on the risk bounds. This effectively decouples the *adversarial* property of features from the *stochastic* property of the noisy labels.

To understand how Theorem 3 operates, we consider the following example:

**Example 4.** *Let $\mathcal{H} \subset \{0, 1\}^{\mathcal{X}}$, and $\mathcal{K}$ be the bounded Bernoulli noise kernel with parameter $\eta$ in Example 1. For any $h_i, h_j \in \mathcal{H}$, we construct the following testing rule. We may assume, w.l.o.g., that $h_i(\mathbf{x}) \neq h_j(\mathbf{x})$ for all $\mathbf{x} \in \mathcal{X}$, since any $\mathbf{x}$ for which $h_i(\mathbf{x}) = h_j(\mathbf{x})$ do not affect our testing. Moreover, by relabeling, we can assume that $h_i(\mathbf{x}) = 0$ and $h_j(\mathbf{x}) = 1$ for all $\mathbf{x} \in \mathcal{X}$. At time step $t$, after observing the noisy labels $\tilde{y}^{t-1}$, we compute $\hat{\mu}_t = \frac{1}{t-1} \sum_{r=1}^{t-1} \tilde{y}_r$. If $\hat{\mu}_t \geq \frac{1}{2}$, the tester predicts $\hat{y}_t = 1$; else, it predicts $\hat{y}_t = 0$. By Azuma's inequality, the probability of making an error at step $t$ is upper bounded by $e^{-(1-2\eta)^2(t-1)/2}$. Thus, for any $n \leq T$, the probability of making any errors after*

step $n$ is upper bounded by $\sum_{t=n}^{\infty} e^{-(1-2\eta)^2(t-1)/2} \leq \frac{e^{-(1-2\eta)^2 n/2}}{(1-2\eta)^2}$. *Taking* $n = \frac{2\log(1/\delta(1-2\eta)^2)}{(1-2\eta)^2}$ *one can upper bound the probability by* $\delta$. *Therefore, the tuple* $(\mathcal{H}, \mathcal{K})$ *is pairwise testable with* $C(\delta) \leq \frac{2\log(1/\delta(1-2\eta)^2)}{(1-2\eta)^2}$. *Invoking Theorem 3, we have:*

$$B^\delta(\mathcal{H}, \mathcal{K}) \leq O\left(\frac{\log|\mathcal{H}|\log(|\mathcal{H}|/\delta(1-2\eta)^2)}{(1-2\eta)^2}\right). \tag{9}$$

Note that the risk bound in (9) recovers the risk in Example 1 up to a logarithmic factor, though it employs a completely different approach (cf. [1]). Moreover, Example 4 provides the key advantage that the risk holds with *high probability* and at a *fast rate*, which is known to be non-trivial for *cumulative* errors (see, e.g., [22, 21]). To our knowledge, it remains unclear whether the approach proposed in [1] admits a high probability guarantee.

## 3.2 Proof of Theorem 2: the *conditional* Le Cam-Birgé Testing

As demonstrated in Section 3.1, the risk of noisy online *classification* can be reduced to the *pairwise testing* of two hypotheses. However, we still need to construct the explicit pairwise testing rules. This section is devoted to providing a generic testing rule for *general* kernels.

Let $h_0$ and $h_1$ be any two hypotheses. We may assume, w.l.o.g., that $h_0(\mathbf{x}) \neq h_1(\mathbf{x})$ for all $\mathbf{x} \in \mathcal{X}$, since the features for which $h_0$ and $h_1$ agree do not affect the testing. We now provide a more compact characterization of the kernel without explicitly referring to true labels. Let $\mathbf{x}^T$ be any realization of features. For any $i \in \{0,1\}$, $t \in [T]$, and kernel $\mathcal{K}$, we write $\mathcal{Q}_i^{\mathbf{x}_t} := \mathcal{K}(\mathbf{x}_t, h_i(\mathbf{x}_t))$.

We define $\mathcal{Q}_0^J$ and $\mathcal{Q}_1^J$ as the sets of all (joint) distributions over $\tilde{\mathcal{Y}}^J$ induced by the kernel upto time step $J$ for $h_0, h_1$, respectively. Equivalently, for $i \in \{0,1\}$, we have $p \in \mathcal{Q}_i^J$ if and only if for all $t \in [J]$ and $\tilde{y}^{t-1} \in \tilde{\mathcal{Y}}^{t-1}$, the *conditional* marginal $p_{\tilde{y}_t|\tilde{y}^{t-1}} \in \mathcal{Q}_i^{\mathbf{x}_t}$.

The pairwise testing of $h_0, h_1$ at time step $J+1$ is then equivalent to the (robust) *hypothesis testing* w.r.t. sets $\mathcal{Q}_0^J$ and $\mathcal{Q}_1^J$. This is typically resolved using Le Cam-Birgé testing [17, Chapter 32.2] if the distributions are of *product* form. However, this does not hold for our purpose, since the distributions in $\mathcal{Q}_i^J$ can have highly correlated marginals. Our main result for addressing this issue is a *conditional* version of Le Cam-Birgé testing, as stated in Theorem 4 below. To the best of our knowledge, this conditional version is novel.

**Theorem 4** (*conditional* Le Cam-Birgé Testing). *Let* $\mathcal{Q}_0^J$ *and* $\mathcal{Q}_1^J$ *be the classes induced by a kernel upto time* $J$ *as defined above. For any* $t \leq J$, *we denote* $\gamma_t = H^2(\mathcal{Q}_0^{\mathbf{x}_t}, \mathcal{Q}_1^{\mathbf{x}_t})$ *and assume that* $\mathcal{Q}_i^{\mathbf{x}_t}$ *is convex for all* $i \in \{0,1\}$. *Then, there exists a testing rule* $\psi : \tilde{\mathcal{Y}}^J \to \{0,1\}$ *such that*

$$\sup_{p \in \mathcal{Q}_0^J, q \in \mathcal{Q}_1^J} \left\{ \Pr_{\tilde{y}^J \sim p}[\psi(\tilde{y}^J) \neq 0] + \Pr_{\tilde{y}^J \sim q}[\psi(\tilde{y}^J) \neq 1] \right\} \leq 2 \prod_{t=1}^{J} (1 - \gamma_t/2) \leq 2 e^{-\frac{1}{2}\sum_{t=1}^{J} \gamma_t}.$$

*Sketch of Proof.* The proof requires a suitable application of the minimax theorem by expressing the testing error as a *linear function* and arguing that the $\mathcal{Q}_i^J$s are convex. The error bound is then controlled by a careful application of the *chain-rule* of Rényi divergence. See Appendix C. □

Theorem 4 immediately implies the following *cumulative* risk bound:

**Proposition 1.** *Let* $h_0, h_1$ *be any hypotheses,* $\mathbf{x}^T$ *be any realization of features and* $\mathcal{Q}_i^T$, $\mathcal{Q}_i^{\mathbf{x}_t}$ *be defined as above with* $\gamma_t = H^2(\mathcal{Q}_0^{\mathbf{x}_t}, \mathcal{Q}_1^{\mathbf{x}_t})$. *Then, there exists a tester* $\hat{y}^T$ *such that for all* $\delta > 0$, $i \in \{0,1\}$ *and* $\tilde{p} \in \mathcal{Q}_i^T$, *w.p.* $\geq 1 - \delta$ *over* $\tilde{y}^T \sim \tilde{p}$, *we have:*

$$\sum_{t=1}^{T} \mathbf{1}\{h_i(\mathbf{x}_t) \neq \hat{y}_t\} \leq \arg\min_n \left\{ n \in \mathbb{N} : \sum_{t=1}^{n} \gamma_t \geq 2\log(2/\delta) \right\}.$$

*Proof.* Let $n^*$ be the minimal number satisfying the RHS. If $t \leq n^*$ (this can be checked at each time step $t$ using only $\mathbf{x}^t$ and $\mathcal{K}$), we predict arbitrarily. If $t \geq n^* + 1$, we use the tester $\psi$ in Theorem 4 with $J = n^*$ to produce an index $\hat{i} \in \{0,1\}$ and make the prediction $h_{\hat{i}}(\mathbf{x}_t)$ for *all* following time steps. That is, we only use the tester at step $n^* + 1$ and reuse the *same* testing result for all following

time steps. By Theorem 4, the probability of making errors after step $n^* + 1$ is upper bounded by $\delta$. Therefore, the cumulative risk is upper bounded by $n^*$ with probability $\geq 1 - \delta$. $\square$

*Proof of Theorem 2.* Let $h_0, h_1 \in \mathcal{H}$ be any two-hypotheses. For any time step $t$ such that $h_0(\mathbf{x}_t) \neq h_1(\mathbf{x}_t)$, we have, by the well-separation condition, that the gap $\gamma_t \geq \gamma_\text{H}$ in Proposition 1. Consider the following testing rule: for any time step $t$ such that $h_1(\mathbf{x}_t) = h_2(\mathbf{x}_t)$, we predict the agreed label; else, we predict the same way as in Proposition 1. Clearly, we only make errors for the second case. Invoking Proposition 1 with $\gamma_t = \gamma_\text{H}$ for all $t \in [T]$, we have $n^* \leq \frac{2\log(2/\delta)}{\gamma_\text{H}}$. Therefore, the tuple $(\mathcal{H}, \mathcal{K})$ is pairwise testable with $C(\delta) = \frac{2\log(2/\delta)}{\gamma_\text{H}}$. The upper bound on *classification* risk then follows by Theorem 3. The lower bound follows by Le Cam's two point method and constructing a hard hypothesis class using an epoch approach. We refer to Appendix D for the complete details. $\square$

**Remark 1.** *Note that our techniques can be easily extended to* infinite classes *using the covering techniques from [1, 22]. Moreover, by applying Proposition 1, our results can be extended to scenarios where the gap parameters $\gamma_t$ are not uniformly bounded, such as in the case of* Tsybakov-type *noise [8], which would lead to risk bounds that scale* sublinearly *with T, in contrast to the* constant *risk in Theorem 2. We leave the details and extensions for a longer manuscript [23].*

## 4 Tighter Bounds for Binary Labels via $L^2$ Gap

We have demonstrated in Theorem 2 that the minimax risk is tightly characterized by the Hellinger gap induced by the kernel. However, the dependency on $\log |\mathcal{H}|$ remains sub-optimal. We show in this section a tight dependency on $\log |\mathcal{H}|$ for classes with *binary* true labels via the $L^2$ gap.

**Theorem 5.** *Let $\mathcal{H} \subset \{0, 1\}^{\mathcal{X}}$ be any* finite *binary valued class, $\mathcal{K}$ be any noisy kernel that is well-separated at scale $\gamma_\text{L}$ w.r.t. the $L^2$-distance [5] (Definition 3). Then, the* expected *minimax risk, as in Definition 1, is upper bounded by:* $\tilde{r}_T(\mathcal{H}, \mathcal{K}) \leq \frac{16\log|\mathcal{H}|}{\gamma_\text{L}}$.

We begin with the following simple geometry fact that is crucial to our proof.

**Lemma 1.** *Let $\mathcal{Q} \subset \mathcal{D}(\tilde{\mathcal{Y}})$ be a convex and closed set, $p$ be a point outside of $\mathcal{Q}$ with $\gamma \overset{\text{def}}{=} \inf_{q \in \mathcal{Q}} L^2(p, q)$. Denote by $q^* \in \mathcal{Q}$ the (unique) point that attains $L^2(p, q^*) = \gamma$. Then for any $q \in \mathcal{Q}$, we have $L^2(q, p) - L^2(q, q^*) \geq L^2(p, q^*) = \gamma$.*

*Proof.* By the *hyperplane separation theorem*, the hyperplane perpendicular to line segment $p - q^*$ at $q^*$ separates $\mathcal{Q}$ and $p$. Therefore, the degree $\theta$ of angle formed by $p - q^* - q$ is greater than $\pi/2$. By the law of cosines, $L^2(q, p) \geq L^2(q, q^*) + L^2(q^*, p) = L^2(q, q^*) + \gamma$. $\square$

Our key idea of proving Theorem 5 is to reduce the robust (noisy) online classification problem to a suitable conditional distribution estimation problem, as discussed next.

**Online conditional distribution estimation.** Let $\mathcal{F} \subset \mathcal{D}(\tilde{\mathcal{Y}})^{\mathcal{X}}$ be a class of functions mapping $\mathcal{X}$ to *distributions* in $\mathcal{D}(\tilde{\mathcal{Y}})$. Online conditional distribution estimation is a game between *Nature* and an *estimator* that follows the following protocol: (1) at each times step $t$, Nature selects some $\mathbf{x}_t \in \mathcal{X}$ and reveals it to the estimator; (2) the estimator then makes an estimation $\hat{p}_t \in \mathcal{D}(\tilde{\mathcal{Y}})$, based on $\mathbf{x}^t, \tilde{y}^{t-1}$; (3) Nature then selects some $\tilde{p}_t \in \mathcal{D}(\tilde{\mathcal{Y}})$, samples $\tilde{y}_t \sim \tilde{p}_t$ and reveals $\tilde{y}_t$ to the estimator. The goal is to find a (deterministic) estimator $\Phi$ that minimizes the *regret*:

$$\text{Reg}_T(\mathcal{F}, \Phi) = \sup_{f \in \mathcal{F}, \mathbf{x}^T \in \mathcal{X}^T} \mathbb{Q}^T \left[ \sum_{t=1}^{T} L(\tilde{p}_t, \hat{p}_t) - L(\tilde{p}_t, f(\mathbf{x}_t)) \right], \tag{10}$$

where $\hat{p}_t = \Phi(\mathbf{x}^t, \tilde{y}^{t-1})$, $\mathbb{Q}^T$ is the operator specified in Definition 1 by setting $\mathcal{Q}_y^{\mathbf{x}} := \mathcal{D}(\tilde{\mathcal{Y}})$ for all $\mathbf{x}, y$, and $L$ is any divergence. We emphasize that the distributions $\tilde{p}^T$ are *not* necessarily realizable by $f$ and are selected completely arbitrarily. This contrasts with the *well-specified* cases employed in [10, 4], and is the key that enables us to handle the *unknown* noisy label distributions.

We now establish the following key technical lemma, see Appendix E for proof.

---

[5]Recall that $L^2(p, q) = ||p - q||_2^2 \overset{\text{def}}{=} \sum_{m=1}^{M}(p[m] - q[m])^2$.

**Lemma 2.** *Let $\mathcal{F} \subset \mathcal{D}(\tilde{\mathcal{Y}})^{\mathcal{X}}$ be a finite distribution-valued function class. Then, for the $L^2$ divergence, there exists an estimator $\Phi$, i.e., the Exponential Weight Average (EWA) algorithm, such that*

$$\mathsf{Reg}_T(\mathcal{F}, \Phi) \leq 4 \log |\mathcal{F}|.$$

*Moreover, estimation $\hat{p}_t$ is a convex combination of $\{f(\mathbf{x}_t) : f \in \mathcal{F}\}$.*

*Proof Sketch of Theorem 5.* We provide the high level ideas and refer to Appendix F for complete details. We define the following *distribution-valued* function class $\mathcal{F}$ using hypothesis class $\mathcal{H}$ and noisy kernel $\mathcal{K}$. For any $\mathbf{x} \in \mathcal{X}$, we denote by $\mathcal{Q}_0^{\mathbf{x}}$ and $\mathcal{Q}_1^{\mathbf{x}}$ the sets of noisy label distributions corresponding to labels 0 and 1, respectively. Since the kernel $\mathcal{K}$ is well-separated at scale $\gamma_{\mathrm{L}}$ under $L^2$ divergence, we have, by the *hyperplane separation theorem*, that there must be $q_0^{\mathbf{x}} \in \mathcal{Q}_0^{\mathbf{x}}$ and $q_1^{\mathbf{x}} \in \mathcal{Q}_1^{\mathbf{x}}$ such that $L^2(q_0^{\mathbf{x}}, q_1^{\mathbf{x}}) = L^2(\mathcal{Q}_0^{\mathbf{x}}, \mathcal{Q}_1^{\mathbf{x}}) \geq \gamma_{\mathrm{L}}$. We now define for any $h \in \mathcal{H}$ the function $f_h$ such that $\forall \mathbf{x} \in \mathcal{X}, \ f_h(\mathbf{x}) = q_{h(\mathbf{x})}^{\mathbf{x}}$. Let $\mathcal{F} = \{f_h : h \in \mathcal{H}\}$ and $\Phi$ be the estimator from Lemma 2 with class $\mathcal{F}$ and $L^2$ divergence (using $\mathbf{x}^T, \tilde{y}^T$ from the *original* noisy classification game). Our *classification* rule is defined as $\hat{y}_t = \arg\min_y \{L^2(q_y^{\mathbf{x}_t}, \hat{p}_t) : y \in \{0, 1\}\}$. That is, we predict the label $y$ so that $q_y^{\mathbf{x}_t}$ is closer to $\hat{p}_t$ under $L^2$ divergence, where $\hat{p}_t = \Phi(\mathbf{x}^t, \tilde{y}^{t-1})$.

Let $h^* \in \mathcal{H}$ be the underlying true classification function. We have by Lemma 2 that

$$\sup_{\mathbf{x}^T \in \mathcal{X}^T} \mathbb{Q}_{\mathcal{K}}^T \left[ \sum_{t=1}^T L^2(\tilde{p}_t, \hat{p}_t) - L^2(\tilde{p}_t, f_{h^*}(\mathbf{x}_t)) \right] \leq 4 \log |\mathcal{F}| \leq 4 \log |\mathcal{H}|, \tag{11}$$

where $\mathbb{Q}_{\mathcal{K}}^T$ is the operator in Definition 1. Now, our key technical goal is to show that $L^2(\tilde{p}_t, \hat{p}_t) - L^2(\tilde{p}_t, f_{h^*}(\mathbf{x}_t)) \geq L^2(\hat{p}_t, f_{h^*}(\mathbf{x}_t)) \geq \frac{\gamma_{\mathrm{L}}}{4} 1\{\hat{y}_t \neq h^*(\mathbf{x}_t)\}$ via Lemma 1 and a *geometric* argument, as illustrated in the figure below:

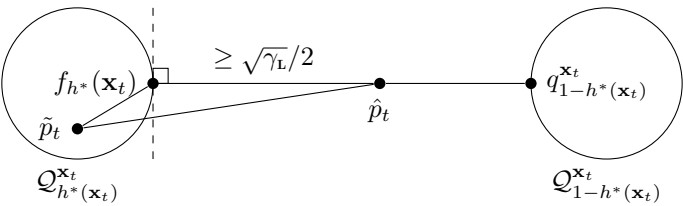

The expected minimax risk bound $\sum_{t=1}^T 1\{\hat{y}_t \neq h^*(\mathbf{x}_t)\} \leq \frac{16 \log |\mathcal{H}|}{\gamma_{\mathrm{L}}}$ then follows from (11). $\qquad\square$

Although both our proofs and those provided in [1] are based on the EWA algorithm, the analysis and resulting algorithms are fundamentally different. For instance, in [1], the learning rate of EWA depends on the parameter $\eta$, while we set it to $1/4$ (see Appendix E). More importantly, our proof applies to *any* noisy kernel that satisfies the well-separation condition (including cases where $|\tilde{\mathcal{Y}}| > 2$), which benefits from our *geometric* interpretation of the kernels. Interestingly, for the specific setting investigated in [1] (i.e., Example 1), our result yields the same order up to a constant factor, since $1 - 2\sqrt{\eta(1-\eta)} = \Theta((1 - 2\eta)^2)$ for $\eta \in [0, \frac{1}{2})$. In general, we have $4\gamma_{\mathrm{L}} \leq \gamma_{\mathrm{H}} \leq \sqrt{M \gamma_{\mathrm{L}}}$.

## 5 Discussion

In this paper, we provide nearly matching lower and upper bounds for online classification with noisy labels via the Hellinger gap of the induced noisy label distributions. Our approach works for a wide range of hypothesis classes and noisy mechanisms. We expect our results to have a wide range of applications, such as online learning under (local) differential privacy constraints and online denoising tasks involving data derived from (noisy) physical measurements (such as learning from quantum data [15]). The main open problem remaining is to close the logarithmic gap in Theorem 2 for *general* kernels. While our work primarily focuses on the information-theoretically achievable minimax risks, we believe that finding computationally efficient predictors (including oracle-efficient methods as in [14]) would also be of significant interest.

## Acknowledgements

This work was partially supported by the NSF Center for Science of Information (CSoI) Grant CCF-0939370, and also by NSF Grants CCF-2006440, CCF-2007238, and CCF-2211423.

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

## A  Martingale Concentration Inequalities

In this appendix, we present some standard concentration results for martingales, which will be useful for deriving high probability guarantees. We refer to [27, Chapter 13.1] for the proofs.

**Lemma 3** (Azuma's Inequality). *Let $X_1, \cdots, X_T$ be an arbitrary random process adaptive to some filtration $\{\mathcal{F}_t\}_{t \leq T}$ such that $|X_t| \leq M$ for all $t \leq T$. Let $Y_t = \mathbb{E}[X_t \mid \mathcal{F}_{t-1}]$ be the conditional expected random variable of $X_t$. Then for all $\delta > 0$, we have*

$$\Pr\left[\sum_{t=1}^{T} Y_t < \sum_{t=1}^{T} X_t + M\sqrt{(T/2)\log(1/\delta)}\right] \geq 1 - \delta,$$

*and*

$$\Pr\left[\sum_{t=1}^{T} Y_t > \sum_{t=1}^{T} X_t - M\sqrt{(T/2)\log(1/\delta)}\right] \geq 1 - \delta.$$

The following lemma provides a tighter concentration when $X_t \geq 0$, which can be viewed as an Martingale version of the multiplicative Chernoff bound.

**Lemma 4** ([27, Theorem 13.5]). *Let $X_1, \cdots, X_T$ be an arbitrary random process adaptive to some filtration $\{\mathcal{F}_t\}_{t \leq T}$ such that $0 \leq X_t \leq M$ for all $t \leq T$. Let $Y_t = \mathbb{E}[X_t \mid \mathcal{F}_{t-1}]$ be the conditional expected random variable of $X_t$. Then for all $\delta > 0$ we have*

$$\Pr\left[\sum_{t=1}^{T} Y_t < 2\sum_{t=1}^{T} X_t + 2M\log(1/\delta)\right] \geq 1 - \delta,$$

*and*

$$\Pr\left[\sum_{t=1}^{T} Y_t > \frac{1}{2}\sum_{t=1}^{T} X_t - (M/2)\log(1/\delta)\right] \geq 1 - \delta.$$

*Proof.* Applying [27, Thm 13.5] with $\xi_t = X_t/M$ and $\lambda = 1$ in the theorem. $\qquad\square$

## B  Proof of Theorem 3

Let $h_{k^*} \in \mathcal{H}$ be the underlying true classification function and $\mathbf{x}^T$ be the realization of features. We take $C = C(\delta/2K)$ in Algorithm 1. By definition of *pairwise testability* and union bound, we have w.p. $\geq 1 - \delta/2$ over the randomness of $\tilde{y}^T$ and the internal randomness of $\Phi_{k,k^*}$s that for all $k \in [K]$,

$$\sum_{t=1}^{T} \mathbf{1}\{h_{k^*}(\mathbf{x}_t) \neq \Phi_{k,k^*}(\mathbf{x}^t, \tilde{y}^{t-1})\} \leq C(\delta/(2K)). \tag{12}$$

Note that for any other $\{i, j\} \not\ni k^*$, equation (12) may not hold for predictor $\Phi_{i,j}$. However, our following argument relies only on the guarantees for predictors $\Phi_{k,k^*}$, which effectively makes our pairwise testing *realizable*.

We now condition on the event defined in (12). Let $\mathbf{v}_t^k$ with $k \in [K]$ and $t \in [T]$ be the *surrogate loss* vector, as defined in (7). We observe the following key properties

    1. We have for all $t \in [T]$ that

$$\max_{j \in [K]} \sum_{r=1}^{t} \mathbf{v}_r^{k^*}[j] \leq C(\delta/(2K)); \tag{13}$$

    2. For any $k \neq k^*$, we have for all $t \in [T]$:

$$\max_{j \in [K]} \sum_{r=1}^{t} \mathbf{v}_r^k[j] \geq \sum_{r=1}^{t} \mathbf{1}\{h_k(\mathbf{x}_r) \neq h_{k^*}(\mathbf{x}_t)\} - C(\delta/(2K)). \tag{14}$$

The first property is straightforward by the definition of $\mathbf{v}_t^k$ and (12). The second property holds since the lower bound is attained when $j = k^*$.

We now analyze the performance of Algorithm 1. By property (13), we know that $k^* \in S^t$ for all $t \in [T]$, i.e., $|S^t| \geq 1$. Let $N_t = |S^t|$. We define for all $t \in [T]$ the *potential*:

$$E_t = \sum_{k \in S^t} \max \left\{ 0, 2C(\delta/(2K)) - \sum_{r=1}^{t} 1\{h_k(\mathbf{x}_r) \neq h_{k^*}(\mathbf{x}_r)\} \right\}.$$

Clearly, we have $E_t \leq 2C(\delta/(2K))N_t$. Let $D_t = |\{k \in S^t : h_k(\mathbf{x}_t) \neq h_{k^*}(\mathbf{x}_t)\}|$. We have:

$$D_t \leq N_t - N_{t+1} + E_t - E_{t+1}, \tag{15}$$

since for any $k \in S_t$ such that $h_k(\mathbf{x}_t) \neq h_{k^*}(\mathbf{x}_t)$, either $k$ is removed from $S^{t+1}$ (which contributes at most $N_t - N_{t+1}$) or its contribution to $E_{t+1}$ is decreased by 1 when compared to $E_t$ (this is because by our construction of Algorithm 1 and property (14) once the contributions of $k$ to $E_t$ equals 0 it must be excluded from $S^{t+1}$). We have, by definition of $\hat{y}_t$, that:

$$\mathbb{E}\left[1\{h_{k^*}(\mathbf{x}_t) \neq \hat{y}_t\}\right] = \frac{D_t}{|S^t|} \leq \frac{N_t - N_{t+1} + E_t - E_{t+1}}{N_t}. \tag{16}$$

By a standard argument [14, Thm 2], we have:

$$\sum_{t=1}^{T} \frac{N_t - N_{t+1}}{N_t} \leq \sum_{t=1}^{T} \left( \frac{1}{N_t} + \frac{1}{N_t - 1} + \cdots + \frac{1}{N_{t+1} + 1} \right)$$

$$\leq \sum_{k=1}^{K} \frac{1}{k} \leq \log K.$$

Moreover, we observe that

$$\sum_{t=1}^{T} \frac{E_t - E_{t+1}}{N_t} \overset{(a)}{\leq} \frac{2C(\delta/(2K))N_1 - E_2}{N_1} + \sum_{t=2}^{T} \frac{E_t - E_{t+1}}{N_t}$$

$$\overset{(b)}{\leq} \frac{2C(\delta/(2K))(N_1 - N_2)}{N_1}$$

$$+ \frac{2C(\delta/(2K))N_2 - E_3}{N_2} + \sum_{t=3}^{T} \frac{E_t - E_{t+1}}{N_t}$$

$$\overset{(c)}{\leq} 2C(\delta/(2K)) \sum_{t=1}^{T} \frac{N_t - N_{t+1}}{N_t}$$

$$\leq 2C(\delta/(2K)) \log K,$$

where $(a)$ and $(b)$ follow by $E_t \leq 2C(\delta/(2K))N_t$ and $N_t \geq N_{t+1}$; $(c)$ follows by repeating the same argument for another $T - 1$ steps.

Therefore, we conclude

$$\mathbb{E}\left[ \sum_{t=1}^{T} 1\{h_{k^*}(\mathbf{x}_t) \neq \hat{y}_t\} \right] \leq (1 + 2C(\delta/(2K))) \log K,$$

where the randomness is on the selection of $\hat{k}_t \sim S^t$. Since our selection of $\hat{k}_t$ are independent (conditioning on $S^t$) for different $t$, and the indicator is bounded by 1 and non-negative, we can invoke Lemma 4 (second part) to obtain a high probability guarantee of confidence $\delta/2$ by introducing an extra $\log(2/\delta)$ additive term. The theorem now follows by a union bound with the event (12).

## C   Proof of Theorem 4

We start with an application of the minimax theorem to hypothesis testing [6].

---

[6]This result was mentioned in [17, Chapter 32.2], without providing a proof.

**Lemma 5.** *Let $\mathcal{P}_0$ and $\mathcal{P}_1$ be two sets of distributions over a finite domain $\Omega$. If $\mathcal{P}_0$ and $\mathcal{P}_1$ are convex under $L_1$ distance (i.e., total variation), then*

$$\min_{\phi \,:\, \Omega \to [0,1]} \sup_{p_0 \in \mathcal{P}_0, p_1 \in \mathcal{P}_1} \{\mathbb{E}_{\omega \sim p_0}[1 - \phi(\omega)] + \mathbb{E}_{\omega \sim p_1}[\phi(\omega)]\} = 1 - \inf_{p_0 \in \mathcal{P}_0, p_1 \in \mathcal{P}_1} \|p_0 - p_1\|_{\mathsf{TV}}.$$

*Moreover, if $\phi^*$ is the function attains minimal, then the tester $\psi^*(\omega) = 1\{\phi^*(\omega) < 0.5\}$ achieves*

$$\sup_{p_0 \in \mathcal{P}_0, p_1 \in \mathcal{P}_1} \{\mathrm{Pr}_{\omega \sim p_0}[\psi^*(\omega) \neq 0] + \mathrm{Pr}_{\omega \sim p_1}[\psi^*(\omega) \neq 1]\} \leq 2(1 - \inf_{p_0 \in \mathcal{P}_0, p_1 \in \mathcal{P}_1} \|p_0 - p_1\|_{\mathsf{TV}}).$$

*Proof.* Observe that the function $\phi$ can be viewed as a vector in $[0,1]^\Omega$. Moreover, the distributions over $\Omega$ can be viewed as vectors in $[0,1]^\Omega$ as well. Therefore, we have $\mathbb{E}_{\omega \sim p_0}[1 - \phi(\omega)] + \mathbb{E}_{\omega \sim p_1}[\phi(\omega)] = \langle p_0, 1 - \phi \rangle + \langle p_1, \phi \rangle$, which is a linear function w.r.t. both $(p_0, p_1)$ and $\phi$. Since the both $\mathcal{P}_0 \times \mathcal{P}_1$ and $[0,1]^\Omega$ are convex and $[0,1]^\Omega$ is compact, we can invoke the minimax theorem [5, Thm 7.1] to obtain

$$\min_{\phi \,:\, \Omega \to [0,1]} \sup_{p_0 \in \mathcal{P}_0, p_1 \in \mathcal{P}_1} \{\mathbb{E}_{\omega \sim p_0}[1 - \phi(\omega)] + \mathbb{E}_{\omega \sim p_1}[\phi(\omega)]\}$$

$$= \sup_{p_0 \in \mathcal{P}_0, p_1 \in \mathcal{P}_1} \min_{\phi \,:\, \Omega \to [0,1]} \{\mathbb{E}_{\omega \sim p_0}[1 - \phi(\omega)] + \mathbb{E}_{\omega \sim p_1}[\phi(\omega)]\}$$

$$= \sup_{p_0 \in \mathcal{P}_0, p_1 \in \mathcal{P}_1} \{1 - \|p_0 - p_1\|_{\mathsf{TV}}\},$$

where the last equality follows by Le Cam's two point lemma [17, Theorem 7.7]. Let $\phi^*$ be the function attains minimal and $\psi^*(\omega) = 1\{\phi^*(\omega) < 0.5\}$. We have $1\{\psi^*(\omega) \neq i\} \leq 2(1 - i - \phi^*(\omega))$ for all $i \in \{0, 1\}$. To see this, for $i = 0$, we have $\psi^*(\omega) \neq 0$ only if $\phi^*(\omega) < 0.5$, thus $1 - \phi^*(\omega) \geq 0.5$ (the case for $i = 1$ follows similarly). Therefore, we have for all $p_0 \in \mathcal{P}_0, p_1 \in \mathcal{P}_1$

$$\mathrm{Pr}_{\omega \sim p_0}[\psi^*(\omega) \neq 0] + \mathrm{Pr}_{\omega \sim p_1}[\psi^*(\omega) \neq 1] \leq 2(\mathbb{E}_{\omega \sim p_0}[1 - \phi^*(\omega)] + \mathbb{E}_{\omega \sim p_1}[\phi^*(\omega)]).$$

This completes the proof. $\qquad\square$

We now establish the following key property, which demonstrates that the distribution classes constructed in Theorem 4 satisfy the condition of Lemma 5.

**Lemma 6.** *Let $\mathcal{Q}_0^J$ and $\mathcal{Q}_1^J$ be the sets in Theorem 4. Then $\mathcal{Q}_0^J$ and $\mathcal{Q}_1^J$ are convex under $L_1$ distance.*

*Proof.* Let $p_1, p_2 \in \mathcal{Q}_i^J$ for $i \in \{0, 1\}$ and $\lambda \in [0, 1]$. We need to show that $p = \lambda p_1 + (1 - \lambda)p_2 \in \mathcal{Q}_i^J$ as well. For any given $t \in [T]$, we have

$$p(\tilde{y}_t \mid \tilde{y}^{t-1}) = \frac{\lambda p_1(\tilde{y}^t) + (1 - \lambda)p_2(\tilde{y}^t)}{\lambda p_1(\tilde{y}^{t-1}) + (1 - \lambda)p_2(\tilde{y}^{t-1})}$$

$$= \lambda \frac{p_1(\tilde{y}^{t-1})}{p(\tilde{y}^{t-1})} p_1(\tilde{y}_t \mid \tilde{y}^{t-1}) + (1 - \lambda) \frac{p_2(\tilde{y}^{t-1})}{p(\tilde{y}^{t-1})} p_2(\tilde{y}_t \mid \tilde{y}^{t-1}) \in \mathcal{Q}_i^{\mathbf{x}_t}$$

where the last inclusion follows by convexity of $\mathcal{Q}_i^{\mathbf{x}_t}$ as assumed in Theorem 4. Therefore, we have $p \in \mathcal{Q}_i^J$ by definition of $\mathcal{Q}_i^J$. $\qquad\square$

Now, our main technical part is to bound the total variation $\mathsf{TV}(\mathcal{Q}_0^J, \mathcal{Q}_1^J)$. The primary challenge comes from controlling the dependencies of conditional marginals of the distributions. To this end, we introduce the concept of *Renyi divergence*. Let $p_1, p_2$ be two distributions over the same finite domain $\Omega$, the $\alpha$-Renyi divergence is defined as

$$D_\alpha(p_1, p_2) = \frac{1}{\alpha - 1} \log \mathbb{E}_{\omega \sim p_2} \left[ \left( \frac{p_1(\omega)}{p_2(\omega)} \right)^\alpha \right].$$

If $p, q$ are distributions over domain $\Omega_1 \times \Omega_2$ and $r$ is a distribution over $\Omega_1$, then the *conditional $\alpha$-Renyi divergence* is defined as

$$D_\alpha(p, q \mid r) = \frac{1}{\alpha - 1} \log \mathbb{E}_{\omega_1 \sim r} \left[ \sum_{\omega_2 \in \Omega_2} p(\omega_2 \mid \omega_1)^\alpha q(\omega_2 \mid \omega_1)^{1-\alpha} \right].$$

The following property about Renyi divergence is well known [17, Chapter 7.12]:

**Lemma 7.** *Let $p, q$ be two distributions over $\Omega_1 \times \Omega_2$ and $p^{(1)}$ and $q^{(1)}$ be the restrictions of $p, q$ on $\Omega_1$, respectively. Then the following chain rule holds*

$$D_\alpha(p, q) = D_\alpha(p^{(1)}, q^{(1)}) + D_\alpha(p, q \mid r),$$

*where $r(\omega_1) = p^{(1)}(\omega_1)^\alpha q^{(1)}(\omega_1)^{1-\alpha} e^{-(\alpha-1)D_\alpha(p^{(1)},q^{(1)})}$ is a distribution over $\Omega_1$.*

The following key result bounds the Renyi divergence between $\mathcal{Q}_0^J$ and $\mathcal{Q}_1^J$:

**Proposition 2.** *Let $\mathcal{Q}_0^J$ and $\mathcal{Q}_1^J$ be the sets in Theorem 4. If $\inf_{p \in \mathcal{Q}_0^{\mathbf{x}_t}, q \in \mathcal{Q}_1^{\mathbf{x}_t}} D_\alpha(p, q) \geq \eta_t$ holds for all $t \leq J$. Then*

$$\inf_{p \in \mathcal{Q}_0^J, q \in \mathcal{Q}_1^J} D_\alpha(p, q) \geq \sum_{t=1}^J \eta_t.$$

*Proof.* We prove by induction on $J$. The base case for $J = 1$ is trivial, since $\mathcal{Q}_0^1 = \mathcal{Q}_0^{\mathbf{x}_1}$ and $\mathcal{Q}_1^1 = \mathcal{Q}_1^{\mathbf{x}_1}$. We now prove the induction step with $J \geq 2$. For any pair $p \in \mathcal{Q}_0^J$ and $q \in \mathcal{Q}_1^J$, we have by Lemma 7 that $D_\alpha(p, q) = D_\alpha(p^{(1)}, q^{(1)}) + D_\alpha(p, q \mid r)$, where $p^{(1)}, q^{(1)}$ are restrictions of $p, q$ on $\tilde{y}^{J-1}$ and $r$ is a distribution over $\tilde{\mathcal{Y}}^{J-1}$. By definition of $\alpha$-Renyi divergence, we have

$$
\begin{aligned}
D_\alpha(p, q \mid r) &\geq \inf_{\tilde{y}^{J-1}} \frac{1}{\alpha - 1} \log \sum_{\tilde{y}_J \in \tilde{\mathcal{Y}}} p(\tilde{y}_J \mid \tilde{y}^{J-1})^\alpha q(\tilde{y}_J \mid \tilde{y}^{J-1})^{1-\alpha} \\
&= \inf_{\tilde{y}^{J-1}} D_\alpha(p_{\tilde{y}_J | \tilde{y}^{J-1}}, q_{\tilde{y}_J | \tilde{y}^{J-1}}) \\
&\overset{(a)}{\geq} \inf_{p \in \mathcal{Q}_0^{\mathbf{x}_J}, q \in \mathcal{Q}_1^{\mathbf{x}_J}} D_\alpha(p, q) \overset{(b)}{\geq} \eta_J,
\end{aligned}
$$

where $(a)$ follows since $p_{\tilde{y}_J | \tilde{y}^{J-1}} \in \mathcal{Q}_0^{\mathbf{x}_J}$ and $q_{\tilde{y}_J | \tilde{y}^{J-1}} \in \mathcal{Q}_1^{\mathbf{x}_J}$ by the definition of $\mathcal{Q}_0^J$ and $\mathcal{Q}_1^J$; $(b)$ follows by assumption. The result then follows by induction hypothesis $D_\alpha(p^{(1)}, q^{(1)}) \geq \sum_{t=1}^{J-1} \eta_t$, since $p^{(1)} \in \mathcal{Q}_0^{J-1}$ and $q^{(1)} \in \mathcal{Q}_1^{J-1}$. $\qquad\square$

The following result converts the Renyi divergence based bounds to that with Hellinger divergence.

**Proposition 3.** *Let $\mathcal{Q}_0^J$ and $\mathcal{Q}_1^J$ be the sets in Theorem 4. If $H^2(\mathcal{Q}_0^{\mathbf{x}_t}, \mathcal{Q}_1^{\mathbf{x}_t}) \geq \gamma_t \geq 0$ holds for all $t \in [J]$. Then*

$$\inf_{p \in \mathcal{Q}_0^J, q \in \mathcal{Q}_1^J} H^2(p, q) \geq 2 \left( 1 - \prod_{t=1}^J (1 - \gamma_t/2) \right).$$

*Proof.* Observe that, for any distributions $p, q$ we have

$$H^2(p, q) = 2(1 - e^{-\frac{1}{2} D_{1/2}(p,q)}). \tag{17}$$

Specifically, for give $p \in \mathcal{Q}_0^J$ and $q \in \mathcal{Q}_1^J$, we have

$$1 - H^2(p, q)/2 = e^{-\frac{1}{2} D_{1/2}(p,q)} \leq e^{-\frac{1}{2} \sum_{t=1}^J \eta_t} = \prod_{t=1}^J e^{-\frac{1}{2} \eta_t} \leq \prod_{t=1}^J (1 - \gamma_t/2),$$

where $\eta_t$s are the constants in Proposition 2 and the last inequality follows by $e^{-\frac{1}{2} \eta_t} \leq 1 - \gamma_t/2$ due to (17) again. This completes the proof. $\qquad\square$

*Proof of Theorem 4.* We have, by Lemma 5, that the testing error is upper bounded by $2(1 - \inf_{p \in \mathcal{Q}_0^J, q \in \mathcal{Q}_1^J} \|p - q\|_{\mathsf{TV}})$. Fix any such $p, q$, we have by [17, Equation 7.22] that $1 - \|p - q\|_{\mathsf{TV}} \leq 1 - \frac{1}{2} H^2(p, q)$. The result then follows by Proposition 3. $\qquad\square$

# D Proof of Theorem 2 (Lower Bound)

We denote $L \leq \log K$ with $K = |\mathcal{H}|$, and $\mathbf{x}_1, \cdots, \mathbf{x}_L$ be $L$ distinct elements in $\mathcal{X}$ satisfies the condition of the theorem. We define for any $\mathbf{b} \in \{0,1\}^L$ a function $h_\mathbf{b}$ such that for all $i \in [L]$, $h_\mathbf{b}(\mathbf{x}_i) = y_i$ if $\mathbf{b}[i] = 0$ and $h_\mathbf{b}(\mathbf{x}_i) = y_i'$ otherwise, where $y_i \neq y_i' \in \mathcal{Y}$ are the elements that satisfy $\inf_{p \in \mathcal{Q}_{y_i}^{\mathbf{x}_i}, q \in \mathcal{Q}_{y_i'}^{\mathbf{x}_i}} \{H^2(p,q)\} \leq \gamma_\mathrm{H}$. Let $\mathcal{H}$ be the class consisting of all such $h_\mathbf{b}$. Let $q_i \in \mathcal{Q}_{y_i}^{\mathbf{x}_i}$ and $q_i' \in \mathcal{Q}_{y_i'}^{\mathbf{x}_i}$ be the elements satisfying $H^2(q_i, q_i') \leq \gamma_\mathrm{H}$. We now partition the features $\mathbf{x}^T$ into $L$ epochs, each of length $T/L$, such that each epoch $i$ has constant feature $\mathbf{x}_i$. Let $\mathbf{h}$ be a random function selected uniformly from $\mathcal{H}$. We claim that for any prediction rule $\hat{y}_t$ and any epoch $i$ we have

$$\mathbb{E}_{\mathbf{h}, \tilde{y}^T} \left[ \sum_{t=iT/L-1}^{(i+1)T/L} \mathbb{1}\{\mathbf{h}(\mathbf{x}_t) \neq \hat{y}_t\} \right] \geq \Omega\left(\frac{1}{\gamma_\mathrm{H}}\right), \tag{18}$$

where $\tilde{y}_t \sim q_i$ if $\mathbf{h}(\mathbf{x}_i) = y_i$ and $\tilde{y}_t \sim q_i'$ otherwise. The proposition now follows by counting the errors for all $L$ epochs.

We now establish (18) using the Le Cam's two point method. Clearly, for each epoch $i$, the prediction performance depends only on the label $\mathbf{y}_i = \mathbf{h}(\mathbf{x}_i)$, which is uniform over $\{y_i, y_i'\}$ and independent for different epochs by construction. For any time step $j$ during the $i$th epoch, we denote by $\tilde{y}^{j-1}$ and $\tilde{y}'^{j-1}$ the samples generated from $q_i$ and $q_i'$, respectively. By the Le Cam's two point method [17, Theorem 7.7] the expected error at step $j$ is lower bounded by

$$\frac{1 - \mathsf{TV}(\tilde{y}^{j-1}, \tilde{y}'^{j-1})}{2} \geq \frac{1 - \sqrt{H^2(\tilde{y}^{j-1}, \tilde{y}'^{j-1})(1 - H^2(\tilde{y}^{j-1}, \tilde{y}'^{j-1})/4)}}{2} \tag{19}$$

where the inequality follows from [17, Equation 7.22]. Note that the RHS of (19) is *monotone decreasing* w.r.t. $H^2(\tilde{y}^{j-1}, \tilde{y}'^{j-1})$, since $H^2(p,q) \leq 2$ for all $p, q$.

By the *tensorization* of Hellinger divergence [17, Equation 7.23], we have

$$H^2(\tilde{y}^{j-1}, \tilde{y}'^{j-1}) = 2 - 2(1 - H^2(q_i, q_i')/2)^{j-1} \leq 2 - 2(1 - \gamma_\mathrm{H}/2)^{j-1},$$

where the last inequality is implied by $H^2(q_i, q_i') \leq \gamma_\mathrm{H}$. Using the fact $\log(1-x) \geq \frac{-x}{1-x}$, we have if $\gamma_\mathrm{H} \leq 1$ and $j - 1 \leq \frac{1}{\gamma_\mathrm{H}}$ then $2 - 2(1 - \gamma_\mathrm{H}/2)^{j-1} \leq 2(1 - e^{-1}) < 2$. Therefore, the RHS of (19) is lower bounded by an *absolute* positive constant for all $j - 1 \leq \frac{1}{\gamma_\mathrm{H}}$, and hence the expected cumulative error will be lower bounded by $\Omega(1/\gamma_\mathrm{H})$ during epoch $i$. This completes the proof.

# E Proof of Lemma 2

Before presenting a formal proof, we first develop some technical concepts. Let $\tilde{\mathcal{Y}}$ be the noisy label set and $\mathcal{D}(\tilde{\mathcal{Y}})$ be the class of distributions over $\tilde{\mathcal{Y}}$. We say a function $\ell : \tilde{\mathcal{Y}} \times \mathcal{D}(\tilde{\mathcal{Y}}) \to \mathbb{R}^+$ is $\alpha$-exp-concave if for any $\tilde{y} \in \tilde{\mathcal{Y}}$, the function $e^{-\alpha \ell(\tilde{y}, p)}$ is concave w.r.t. $p$ for some $\alpha \in \mathbb{R}^{\geq 0}$.

**Proposition 4.** *The function $\ell(\tilde{y}, p) = ||e_{\tilde{y}} - p||_2^2$ is $1/4$-Exp-concave, where $e_{\tilde{y}}$ denotes distribution assigning probability $1$ on $\tilde{y}$.*

*Proof.* We have by [13, Lemma 4.2] that a function $f$ is $\alpha$-Exp-concave if and only if

$$\alpha \nabla f(p) \nabla f(p)^\mathsf{T} \preceq \nabla^2 f(p).$$

For any $q \in \mathcal{D}(\tilde{\mathcal{Y}})$, we denote $f(p) = ||p - q||_2^2$. We have $\nabla f(p) = 2(p - q)$ and $\nabla^2 f(p) = 2I$, where $I$ is the identity matrix. Taking any $u \in \mathbb{R}^J$, we have $\frac{1}{4}\langle u, 2(p-q)\rangle^2 \leq ||u||_2^2 ||p - q||_2^2 \leq 2||u||_2^2 = 2u^\mathsf{T} I u$, where the first inequality follows by Cauchy-Schwarz inequality and the second inequality follows by:

$$||p - q||_2^2 = \sum_{\tilde{y} \in \tilde{\mathcal{Y}}} (p[\tilde{y}] - q[\tilde{y}])^2 \leq \sum_{\tilde{y} \in \tilde{\mathcal{Y}}} \max\{p[\tilde{y}], q[\tilde{y}]\}^2 \leq \sum_{\tilde{y} \in \tilde{\mathcal{Y}}} p[\tilde{y}]^2 + q[\tilde{y}]^2 \leq 2,$$

since $p, q \in \mathcal{D}(\tilde{\mathcal{Y}})$. This completes the proof. $\qquad\square$

We now introduce the *Exponential Weighted Average (EWA)* algorithm and its regret analysis under the Exp-concave losses, which is mostly standard [5, Chapter 3.3] and we include it here for completeness. Let $\mathcal{F} = \{f_1, \cdots, f_K\} \subset \mathcal{D}(\tilde{\mathcal{Y}})^{\mathcal{X}}$ be a $\mathcal{D}(\tilde{\mathcal{Y}})$-valued function class and $\ell : \tilde{\mathcal{Y}} \times \mathcal{D}(\tilde{\mathcal{Y}}) \to \mathbb{R}^{\geq 0}$ be $\alpha$-Exp-concave. The EWA algorithm is presented in Algorithm 2.

---

**Algorithm 2:** Exponential Weighted Average (EWA) estimator

---

**Input**: Class $\mathcal{F} = \{f_1, \cdots, f_K\}$ and $\alpha$-Exp-concave loss $\ell$

Set $\mathbf{w}^1 = \{1, \cdots, 1\} \in \mathbb{R}^K$;

**for** $t = 1, \cdots, T$ **do**

    Receive $\mathbf{x}_t$;

    Make prediction:

$$\hat{p}_t = \frac{\sum_{k=1}^K \mathbf{w}^t[k] f_k(\mathbf{x}_t)}{\sum_{k=1}^K \mathbf{w}^t[k]}.$$

    Receive noisy label $\tilde{y}_t$;

    **for** $k \in [K]$ **do**

        Set $\mathbf{w}^{t+1}[k] = \mathbf{w}^t[k] e^{-\alpha \ell(\tilde{y}_t, f_k(\mathbf{x}_t))}$;

---

Algorithm 2 provides the following regret bound:

**Proposition 5** ([5, Proposition 3.1]). *Let $\mathcal{F} \subset \mathcal{D}(\tilde{\mathcal{Y}})^{\mathcal{X}}$ be any finite class and $\ell$ be an $\alpha$-Exp-concave loss. If $\hat{p}_t$ is the estimator in Algorithm 2, then for any $\mathbf{x}^T \in \mathcal{X}^T$ and $\tilde{y}^T \in \tilde{\mathcal{Y}}^T$ we have*

$$\sup_{f \in \mathcal{F}} \sum_{t=1}^T \ell(\tilde{y}_t, \hat{p}_t) - \ell(\tilde{y}_t, f(\mathbf{x}_t)) \leq \frac{\log |\mathcal{F}|}{\alpha}.$$

*Proof of Lemma 2.* Let $\Phi$ be the EWA estimator as in Algorithm 2 with input class $\mathcal{F}$, loss $\ell(\tilde{y}, p) \stackrel{\text{def}}{=} L^2(e_{\tilde{y}}, p)$ and $\alpha = 1/4$. Let $\tilde{y}^T$ be any realization of the noisy labels. We denote $e_t$ as the standard base of $\mathbb{R}^M$ with value 1 at position $\tilde{y}_t$ and zeros otherwise. By $1/4$-Exp-concavity of loss $\ell$ (Proposition 4) and the regret bound from Proposition 5, we have:

$$\sup_{f \in \mathcal{F}, \mathbf{x}^T \in \mathcal{X}^T, \tilde{y}^T \in \tilde{\mathcal{Y}}^T} \sum_{t=1}^T L^2(e_t, \hat{p}_t) - L^2(e_t, f(\mathbf{x}_t)) \leq 4 \log |\mathcal{F}|. \tag{20}$$

Note that, this bound holds *point-wise* w.r.t. any individual $\mathbf{x}^T$ and $\tilde{y}^T$.

Fix any $\mathbf{x}^T$ and (joint) distribution $\tilde{p}$ over $\tilde{\mathcal{Y}}^T$. We denote $\mathbb{E}_t$ as the conditional expectation on $\tilde{y}_t$ over the randomness of $\tilde{y}^T \sim \tilde{p}$ conditioning on $\tilde{y}^{t-1}$ and denote $\tilde{p}_t$ as the *conditional* marginal. By the elementary identity $\mathbb{E}[L^2(X, p) - L^2(X, q)] = L^2(\mathbb{E}[X], p) - L^2(\mathbb{E}[X], q)$ for any random variable $X$ over $\mathcal{D}(\tilde{\mathcal{Y}})$, we have for all $t \in [T]$ that:

$$\mathbb{E}_t \left[ L^2(e_t, \hat{p}_t) - L^2(e_t, f(\mathbf{x}_t)) \right] = L^2(\tilde{p}_t, \hat{p}_t) - L^2(\tilde{p}_t, f(\mathbf{x}_t)),$$

since $\mathbb{E}_t[e_t] = \tilde{p}_t$ for $\tilde{y}_t \sim \tilde{p}_t$ and $\hat{p}_t$ depends only on $\tilde{y}^{t-1}$. We now take $\mathbb{E}_{\tilde{y}^T}$ on both sides of (20). By $\sup \mathbb{E} \leq \mathbb{E} \sup$ and the law of total probability (i.e., $\mathbb{E}_{\tilde{y}^T}[X_1 + \cdots + X_T] = \mathbb{E}_{\tilde{y}^T}[\mathbb{E}_1[X_1] + \cdots + \mathbb{E}_T[X_T]]$ for any random variables $X^T$), we have:

$$\sup_{f \in \mathcal{F}, \mathbf{x}^T \in \mathcal{X}^T} \sup_{\tilde{p}} \mathbb{E}_{\tilde{y}^T \sim \tilde{p}} \left[ \sum_{t=1}^T L^2(\tilde{p}_t, \hat{p}_t) - L^2(\tilde{p}_t, f(\mathbf{x}_t)) \right] \leq 4 \log |\mathcal{F}|,$$

where $\tilde{p}$ runs over all (joint) distributions over $\tilde{\mathcal{Y}}^T$. The lemma then follows by the equivalence between operators $\mathbb{Q}_{\mathcal{K}}^T \equiv \sup_{\tilde{p}} \mathbb{E}_{\tilde{y}^T}$ when taking the kernel set $\mathcal{Q}_y^{\mathbf{x}} := \mathcal{D}(\tilde{\mathcal{Y}})$ for all $\mathbf{x}, y$ (see the discussion following Definition 1). The last part follows by the fact that the EWA estimator automatically ensures $\hat{p}_t$ is a convex combination of $\{f(\mathbf{x}_t) : f \in \mathcal{F}\}$ for all $t \in [T]$. □

# F Proof of Theorem 5

We define the following distribution valued function class $\mathcal{F}$ using hypothesis class $\mathcal{H}$ and noisy kernel $\mathcal{K}$. For any $\mathbf{x} \in \mathcal{X}$, we denote by $\mathcal{Q}_0^\mathbf{x}$ and $\mathcal{Q}_1^\mathbf{x}$ the sets of noisy label distributions corresponding to labels 0 and 1, respectively. Since the kernel $\mathcal{K}$ is well-separated at scale $\gamma_\mathrm{L}$ under $L^2$ divergence, we have, by the *hyperplane separation theorem*, that there must be $q_0^\mathbf{x} \in \mathcal{Q}_0^\mathbf{x}$ and $q_1^\mathbf{x} \in \mathcal{Q}_1^\mathbf{x}$ such that $L^2(q_0^\mathbf{x}, q_1^\mathbf{x}) = L^2(\mathcal{Q}_0^\mathbf{x}, \mathcal{Q}_1^\mathbf{x}) \geq \gamma_\mathrm{L}$. We now define for any $h \in \mathcal{H}$ the function $f_h$ such that $\forall \mathbf{x} \in \mathcal{X}, \ f_h(\mathbf{x}) = q_{h(\mathbf{x})}^\mathbf{x}$. Let $\mathcal{F} = \{f_h : h \in \mathcal{H}\}$ and $\Phi$ be the estimator from Lemma 2 with class $\mathcal{F}$ and $L^2$ divergence (using $\mathbf{x}^T, \tilde{y}^T$ from the *original* noisy classification game). Our *classification* predictor is as follows:

$$\hat{y}_t = \arg\min_y \{L^2(q_y^{\mathbf{x}_t}, \hat{p}_t) : y \in \{0, 1\}\}. \tag{21}$$

That is, we predict the label $y$ so that $q_y^{\mathbf{x}_t}$ is closer to $\hat{p}_t$ under $L^2$ divergence, where $\hat{p}_t = \Phi(\mathbf{x}^t, \tilde{y}^{t-1})$.

Let $h^* \in \mathcal{H}$ be the underlying true classification function and $\mathbf{x}^T$ be the realization of features. We have by Lemma 2 and $1/4$-Exp-concavity of $L^2$ divergence that [7]

$$\mathbb{Q}_\mathcal{K}^T \left[ \sum_{t=1}^T L^2(\tilde{p}_t, \hat{p}_t) - L^2(\tilde{p}_t, f_{h^*}(\mathbf{x}_t)) \right] \leq 4 \log |\mathcal{F}|, \tag{22}$$

where $\mathbb{Q}_\mathcal{K}^T$ is the operator in Definition 1.

For any time step $t$, we denote by $y_t = h^*(\mathbf{x}_t)$ the true label. Since $q_y^{\mathbf{x}_t} \in \mathcal{Q}_y^{\mathbf{x}_t}$ are the elements satisfying $L^2(q_0^{\mathbf{x}_t}, q_1^{\mathbf{x}_t}) = L^2(\mathcal{Q}_0^{\mathbf{x}_1}, \mathcal{Q}_1^{\mathbf{x}_t}) \geq \gamma_\mathrm{L}$ and $\hat{q}_t$ is a *convex* combination of $q_0^{\mathbf{x}_t}$ and $q_1^{\mathbf{x}_t}$ (Lemma 2), we have $q_{y_t}^{\mathbf{x}_t}$ is the closest element in $\mathcal{Q}_{y_t}^{\mathbf{x}_t}$ to $\hat{p}_t$ under $L^2$ divergence. Note that, we also have $\tilde{p}_t \in \mathcal{Q}_{y_t}^{\mathbf{x}_t}$. Invoking Lemma 1, we find

$$L^2(\tilde{p}_t, \hat{p}_t) - L^2(\tilde{p}_t, q_{y_t}^{\mathbf{x}_t}) \geq L^2(\hat{p}_t, q_{y_t}^{\mathbf{x}_t}). \tag{23}$$

Denote $a_t = L^2(\tilde{p}_t, \hat{p}_t) - L^2(\tilde{p}_t, f_{h^*}(\mathbf{x}_t))$. We have, by (23) and $f_{h^*}(\mathbf{x}_t) = q_{y_t}^{\mathbf{x}_t}$ that $a_t \geq L^2(\hat{p}_t, f_{h^*}(\mathbf{x}_t))$. Therefore:

1. For all $t \in [T]$, $a_t \geq 0$, since $\forall p, q, \ L^2(p, q) \geq 0$;

2. If $\hat{y}_t \neq y_t$, then $a_t \geq \gamma_\mathrm{L}/4$. This is because the event $\{\hat{y}_t \neq y_t\}$ implies that $L^2(\hat{p}_t, q_{y_t}^{\mathbf{x}_t}) \geq L^2(\hat{p}_t, q_{1-y_t}^{\mathbf{x}_t})$. Hence, $L^2(\hat{p}_t, f_{h^*}(\mathbf{x}_t)) = L^2(\hat{p}_t, q_{y_t}^{\mathbf{x}_t}) \geq \gamma_\mathrm{L}/4$. Here, we used the following geometric fact:

$$2\sqrt{L^2(\hat{p}_t, q_{y_t}^{\mathbf{x}_t})} \geq \sqrt{L^2(\hat{p}_t, q_{y_t}^{\mathbf{x}_t})} + \sqrt{L^2(\hat{p}_t, q_{1-y_t}^{\mathbf{x}_t})}$$
$$= \sqrt{L^2(q_{y_t}^{\mathbf{x}_t}, q_{1-y_t}^{\mathbf{x}_t})} \geq \sqrt{\gamma_\mathrm{L}}.$$

This implies that $\forall t \in [T], \ a_t \geq \frac{\gamma_\mathrm{L}}{4} \mathbb{1}\{\hat{y}_t \neq y_t\}$, therefore:

$$\sum_{t=1}^T \mathbb{1}\{\hat{y}_t \neq y_t\} \leq \frac{4}{\gamma_\mathrm{L}} \sum_{t=1}^T L^2(\tilde{p}_t, \hat{p}_t) - L^2(\tilde{p}_t, f_{h^*}(\mathbf{x}_t)).$$

The expected minimax risk now follows from (22).

# G Tight Bounds for Kernel Sets of Size One

In this appendix, we establish an upper bound for the case when the kernel set size $|\mathcal{Q}_y^\mathbf{x}| = 1$ for all $\mathbf{x}, y$. This includes, for instance, the case when the parameter $\eta_t$ is *known* in Example 1.

**Theorem 6.** *Let $\mathcal{H} \subset \mathcal{Y}^\mathcal{X}$ be any finite class and $\mathcal{K}$ be any noisy kernel that is well-separated at scale $\gamma_\mathrm{H}$ w.r.t. squared Hellinger distance such that $|\mathcal{Q}_y^\mathbf{x}| = 1$ for all $\mathbf{x}, y$. Then the high probability minimax risk at confidence $\delta > 0$ is upper bounded by*

$$B^\delta(\mathcal{H}, \mathcal{K}) \leq O\left(\frac{\log(|\mathcal{H}|/\delta)}{\gamma_\mathrm{H}}\right).$$

---

[7]Since $\mathbb{Q}_\mathcal{K}^T[F(\tilde{y}^T)] \leq \mathbb{Q}^T[F(\tilde{y}^T)]$ for any kernel $\mathcal{K}$ and function $F$, where $\mathbb{Q}^T$ is the *unconstrained* operator in (10).

*Proof.* Our proof follows a similar path as in the proof of Theorem 5, but replacing the $L^2$ loss with log-loss. Specifically, for any $h \in \mathcal{H}$, we define $f_h(\mathbf{x}) = q^{\mathbf{x}}_{h(\mathbf{x})}$, where $q^{\mathbf{x}}_{h(\mathbf{x})}$ is the unique element in $\mathcal{Q}^{\mathbf{x}}_{h(\mathbf{x})}$. Denote $\mathcal{F} = \{f_h : h \in \mathcal{H}\}$. We run the EWA algorithm (Algorithm 2) over $\mathcal{F}$ with $\alpha = 1$ and $\ell$ being the log-loss [10], and produce an estimator $\hat{p}^T$. The classifier is then given by

$$\hat{y}_t = \arg\min_{y \in \mathcal{Y}} \{H^2(q^{\mathbf{x}_t}_y, \hat{p}_t)\}.$$

Now, our key observation is that the noisy label distribution $\tilde{p}_t = f_{h^*}(\mathbf{x}_t)$ is *well-specified* (since $|\mathcal{Q}^{\mathbf{x}}_y| = 1$, the only choice for $\tilde{p}_t$ is $f_{h^*(\mathbf{x}_t)}$), where $h^*$ is the ground truth classifier. Therefore, invoking [10, Lemma A.14], we find

$$\Pr\left[\sum_{t=1}^{T} H^2(\tilde{p}_t, \hat{p}_t) \leq \log|\mathcal{F}| + 2\log(1/\delta)\right] \geq 1 - \delta.$$

We claim that $1\{\hat{y}_t \neq h^*(\mathbf{x}_t)\} \leq \frac{4}{\gamma_{\text{H}}} H^2(\tilde{p}_t, \hat{p}_t)$. Clearly, this automatically satisfies if $\hat{y}_t = h^*(\mathbf{x}_t)$. For $\hat{y}_t \neq h^*(\mathbf{x}_t)$, we have $H^2(q^{\mathbf{x}_t}_{\hat{y}_t}, \hat{p}_t) \leq H^2(q^{\mathbf{x}_t}_{h^*(\mathbf{x}_t)}, \hat{p}_t) = H^2(\tilde{p}_t, \hat{p}_t)$ by definition of $\hat{y}_t$. This implies that

$$H^2(\tilde{p}, \hat{p}_t) \geq \frac{1}{4} H^2(q^{\mathbf{x}_t}_{\hat{y}_t}, q^{\mathbf{x}_t}_{h^*(\mathbf{x}_t)}) \geq \frac{\gamma_{\text{H}}}{4},$$

where the first inequality follows by triangle inequality of Hellinger distance (the factor $\frac{1}{4}$ comes from the conversion form squared Hellinger distance to Hellinger distance), and the second inequality follows by definition of $\gamma_{\text{H}}$. Therefore, we have w.p. $\geq 1 - \delta$ that

$$\sum_{t=1}^{T} 1\{\hat{y}_t \neq h^*(\mathbf{x}_t)\} \leq \frac{4}{\gamma_{\text{H}}}(\log|\mathcal{F}| + 2\log(1/\delta)).$$

This completes the proof since $|\mathcal{H}| \geq |\mathcal{F}|$. $\qquad\square$

Observe that the key ingredient in the proof of Theorem 6 is the realizability of $\tilde{p}_t$ by $f_{h^*}$ (i.e., well-specified) due to the property $|\mathcal{Q}^{\mathbf{x}}_y| = 1$, which does not hold for general kernels.

## H  Adaptive v.s. Oblivious Adversaries

In this appendix, we explain how the guarantees for oblivious adversaries can be extended to adaptive adversaries. This primarily follows from [5, Lemma 4.1], but needs careful adaptation to fit our needs. We consider the following abstract treatment: we assume that the adversary performs any *operation* $\mathbb{Q}_t$ at time step $t$ and produces an action $\mathbf{z}_t$. For any randomized prediction rule $\hat{y}^T$, the adaptive risk can be expressed as

$$\mathbb{Q}_1 \mathbb{E}_{\hat{y}_1} \cdots \mathbb{Q}_T \mathbb{E}_{\hat{y}_T} \left[\sum_{t=1}^{T} \ell(\mathbf{z}_t, \hat{y}_t)\right].$$

Assume now that the randomness of $\hat{y}_t$'s is *independent* and that $\hat{y}_t$ depends only on $\mathbf{z}^t$. We claim that

$$\mathbb{Q}_1 \mathbb{E}_{\hat{y}_1} \cdots \mathbb{Q}_T \mathbb{E}_{\hat{y}_T} \left[\sum_{t=1}^{T} \ell(\mathbf{z}_t, \hat{y}_t)\right] = \mathbb{Q}^T \mathbb{E}_{\hat{y}^T} \left[\sum_{t=1}^{T} \ell(\mathbf{z}_t, \hat{y}_t)\right].$$

We prove the case for $T = 2$ to demonstrate the ideas; the general case follows by induction. Observe that

$$
\begin{aligned}
\mathbb{Q}_1 \mathbb{E}_{\hat{y}_1} \mathbb{Q}_2 \mathbb{E}_{\hat{y}_2}[\ell(\mathbf{z}_1, \hat{y}_1) + \ell(\mathbf{z}_2, \hat{y}_2)] &\overset{(a)}{=} \mathbb{Q}_1 \mathbb{E}_{\hat{y}_1}[\ell(\mathbf{z}_1, \hat{y}_1) + \mathbb{Q}_2 \mathbb{E}_{\hat{y}_2} \ell(\mathbf{z}_2, \hat{y}_2)] \\
&\overset{(b)}{=} \mathbb{Q}_1[\mathbb{E}_{\hat{y}_1}[\ell(\mathbf{z}_1, \hat{y}_1)] + \mathbb{E}_{\hat{y}_1} \mathbb{Q}_2 \mathbb{E}_{\hat{y}_2} \ell(\mathbf{z}_2, \hat{y}_2)] \\
&\overset{(c)}{=} \mathbb{Q}_1[\mathbb{E}_{\hat{y}_1}[\ell(\mathbf{z}_1, \hat{y}_1)] + \mathbb{Q}_2 \mathbb{E}_{\hat{y}_2} \ell(\mathbf{z}_2, \hat{y}_2)] \\
&\overset{(d)}{=} \mathbb{Q}_1 \mathbb{Q}_2[\mathbb{E}_{\hat{y}_1}[\ell(\mathbf{z}_1, \hat{y}_1)] + \mathbb{E}_{\hat{y}_2}[\ell(\mathbf{z}_2, \hat{y}_2)]] \\
&\overset{(e)}{=} \mathbb{Q}_1 \mathbb{Q}_2 \mathbb{E}_{\hat{y}_1} \mathbb{E}_{\hat{y}_2}[\ell(\mathbf{z}_1, \hat{y}_1) + \ell(\mathbf{z}_2, \hat{y}_2)]
\end{aligned}
$$

where $(a)$ follows since $\ell(\mathbf{z}_1, \hat{y}_1)$ is independent of $\mathbb{Q}_2 \mathbb{E}_{\hat{y}_2}$; $(b)$ follows by the linearity of expectation; $(c)$ follows by the *independence* of $\hat{y}_1$ and $\hat{y}_2$, since the term $\mathbb{Q}_2 \mathbb{E}_{\hat{y}_2} \ell(\mathbf{z}_2, \hat{y}_2)$ has nothing to do with the realization of $\hat{y}_1$; $(d)$ follows since $\mathbb{E}_{\hat{y}_1}[\ell(\mathbf{z}_1, \hat{y}_1)]$ is independent of $\mathbf{z}_2$; $(e)$ follows by the linearity of expectation.

Observe that, all the predictors constructed in this paper have *independent* internal randomness (in fact, the only place where randomness is introduced is in Algorithm 1); thus, our derived risk bounds hold for adaptive adversaries as well.

