# OpenReview forum: "Information-theoretic Limits of Online Classification with Noisy Labels"
_NeurIPS.cc/2024/Conference — NeurIPS 2024 poster_

### Official Review · Reviewer_uBfk · 2024-07-05

**Soundness:** 2
**Presentation:** 3
**Contribution:** 2
**Rating:** 4
**Confidence:** 3

**Summary:**

The paper addresses the problem of online classification where the true labels are corrupted by unknown stochastic noise and the features are adversarially generated. The main contributions of this paper include: 1. Establishing fundamental limits of minimax risk in online classification with noisy labels by nearly matching lower and upper bounds across various hypothesis classes and noisy kernels; 2. Introducing a reduction to an online comparison scheme of two hypotheses and a new conditional version of Le Cam-Birgé testing suitable for online settings; 3. Characterizing the minimax risk using the Hellinger gap of noisy label distributions.

**Strengths:**

+ The theoretical analysis is rigorous, providing clear bounds and demonstrating their tightness through detailed proofs. The paper systematically addresses various aspects of the problem, ensuring a comprehensive exploration.
+ The paper is well-structured, with clear definitions and thorough explanations of the concepts and results. Theorems and proofs are presented in a logical sequence.

**Weaknesses:**

- While the theoretical contributions are good, the paper could benefit from discussing practical implementations and real-world applications of the proposed methods.
- Some assumptions, such as the nature of the noise and the hypothesis class, might be restrictive. A more detailed examination of these assumptions' impact on the results' generality would strengthen the paper.
- The paper lacks experimental validation of the theoretical results. Providing empirical evidence, even in simulated settings, would enhance the credibility and applicability of the findings.

**Questions:**

1. How do the assumptions regarding the noise model and hypothesis class impact the generality of the results? Can the methods be adapted to other types of noise or broader hypothesis classes?
2. Have you considered any practical applications or scenarios where your theoretical findings could be applied? If so, what are the potential challenges in these applications?

**Limitations:**

- Limitations: In the discussion section, the discussion on the limitations is not sufficient. It's difficult for readers to find this part.
- Societal Impact: As this is a pure theory paper, there are no direct societal impacts. However, the authors could briefly mention potential positive impacts, such as improved robustness in machine learning models dealing with noisy data, and any negative implications if misapplied, such as in adversarial settings.

---

> ### Author Rebuttal · Authors · 2024-07-31
>
> We thank the reviewer for the detailed review and helpful comments. We address the main concerns raised below:
>
> **Real-World Application:** Our theoretical work is motivated by a number of important real-world applications. These applications involve, e.g., learning under measurement imprecision, estimation and communication error, and noise introduced for differential privacy. For instance, in a differential privacy setting, noise can arise from certain privacy mechanisms, and our noise kernel for a given label corresponds to the collection of all distributions when passed through the privacy mechanism. Additionally, in quantum learning, one may consider the task of classifying quantum states into certain classes. Here, the kernel set for any given label is the collection of all distributions induced by measurements on the quantum states corresponding to that class. Finally, in the context of genome sequencing—where the raw output from the sequencer must be classified into one of four bases (A, T, C, G)—this is done using a base-calling process that is inherently noisy. These problems can be cast within our framework. We refer the reviewer to our global rebuttal for a concrete construction that demonstrates how our theory applies in the context of differential private online learning.
>
> **Assumptions:** We would like to clarify that our work is not intended to introduce any specific noise models. Rather, our main focus is to provide a *characterization* to determine, for any given noise model, the *best achievable* minimax risk. Since our formulation of noise kernels is *general* and we do not make structural assumptions on the class (except finiteness), our result is applicable to a wide range of noisy online classification problems (such as those highlighted above). Although our results are stated for uniformly bounded gaps and finite classes, they can be extended to *non-uniform* gaps and *infinite* classes (see our global rebuttal and responses to Reviewers JscE and crkZ).
>
> **Experiments:** Our work is primarily focused on theoretical aspects (particularly the *worst-case* minimax risk), and experiments are typically not expected for such pure theory contributions in the literature. For example, Ben-David et al. (2009) and many other foundational works in our references do not include experiments. Our results are established via rigorous proofs, ensuring that the worst-case risk bounds will hold for real data, provided the noisy data follow the prescribed kernel. While empirical validation is valuable, it lies beyond the scope of our current theoretical investigation and could be a potential avenue for future work to complement our findings.
>
> **Limitations:** We appreciate the reviewer's feedback regarding the discussion of limitations. We would be grateful if the reviewer could specify any particular limitations, and we will be happy to address them.

---

> > ### Comment · Reviewer_uBfk · 2024-08-13
> >
> > Real-World Applications: While it is appreciated that you have highlighted potential applications of your theoretical framework in areas such as differential privacy, these examples remain largely theoretical. Specific details on how your model can be practically implemented in these scenarios are still lacking. For instance, in the context of differential privacy, how does your model handle varied levels of noise introduced by different privacy-preserving mechanisms? Additionally, the practical challenges of adapting your theoretical model to real-world data constraints in these fields should be thoroughly discussed.
> >
> > Assumptions: The clarification on the generality of your noise model is helpful. However, the impact of these assumptions on the robustness and adaptability of the model could benefit from a deeper exploration.
> >
> > Experiments: It is understood that your paper focuses on theoretical aspects; however, including some experimental validation, even if preliminary, would significantly strengthen your paper's impact and credibility. Theoretical proofs, while rigorous, often benefit from empirical demonstrations to highlight their applicability and to verify theoretical predictions under practical conditions. It could be beneficial to include simulations or synthetic experiments that demonstrate the theoretical concepts and provide insight into their performance in controlled settings.
> >
> > Limitations: While the global rebuttal may address broader concerns, the specific discussion in the paper about limitations, particularly how they might affect the interpretation and application of your results, seems insufficient. A more detailed examination of potential pitfalls or the conditions under which your proposed work may not perform as expected would provide a more balanced view and help practitioners gauge the utility of your findings.
> >
> > Overall, most concerns have NOT been addressed yet. Addressing the above points could further enhance the work.

---

> > > ### Author Response · Authors · 2024-08-13
> > >
> > > We thank the reviewer for the response.
> > >
> > > However, we respectfully disagree with the reviewer's assessment of our paper. Our work is theoretical in nature and is labeled as "Learning Theory." Therefore, its merit should primarily be judged based on the theoretical contributions. We believe it is not fair to reject a theoretical paper solely for lacking practical implementation, considering that many purely theoretical papers have been published at NeurIPS.
> > >
> > > **Practical Implementation:** The purpose of our paper is to establish the theoretical foundations of online classification under general noise models. Our work is not tailored to specific noise models but provides a general *analysis framework* that can be applied to specific noise model at hand. Our work will inspire practitioners to develop practical algorithms using the algorithmic ideas proposed here. However, we believe that placing too much emphasis on practical implementation in specific application scenarios would detract from the main message of the paper and could be itself a separate paper.
> > >
> > > **Assumptions:** To our knowledge, the only "assumption" we made is that the real data follows the prescribed noisy kernel, which is entirely determined by the noise model at hand. Our theoretical results clearly state the conditions on the kernel under which the results hold. If one has to say something, perhaps the adversarial assumption may be pessimistic, so that for real data, better performance might be achievable than our risk bound predicts. However, this does not affect the applicability of our algorithm to such tasks. We would greatly appreciate it if the reviewer could specify any particular assumptions the reviewer feels need further discussion.
> > >
> > > We now address the specific comments made by the reviewer:
> > >
> > > > How does your model handle varied levels of noise introduced by different privacy-preserving mechanisms?
> > >
> > > We note that the parameter $\eta$ is only an upper bound on the noise level. Our algorithm automatically accommodates scenarios where each local party chooses varying noise parameters that is upper bounded by $\eta$, as our setting allows the noisy label distribution to be selected adversarially.
> > >
> > > > A more detailed examination of potential pitfalls or the conditions under which your proposed work may not perform as expected would provide a more balanced view and help practitioners gauge the utility of your findings.
> > >
> > > As we pointed out above, our work provides a *framework* to analyze general noise models. To ensure our theory applies, one must ensures that the real data follows the noisy kernel. For instance, in differential privacy, the kernel is designed and therefore fully controlled by the learner. In other cases, such as noisy communication, the kernel can typically be estimated. To our knowledge, all the necessary conditions of our theoretical results have been clearly stated in our work.

---

### Official Review · Reviewer_GopC · 2024-07-06

**Soundness:** 3
**Presentation:** 3
**Contribution:** 3
**Rating:** 6
**Confidence:** 3

**Summary:**

This paper studies online classification with stochastic label noise, where a hypothesis $h^*$ is drawn from some hypothesis class $H$ and the learner wants to learn $h^*$ by observing the noisy labels of the online data.
The standard mistake bound used in online learning counts the number of predictions different from the true labels. In this paper, instead, the mistake bound counts the number of predictions that disagree with the target $h^*$.
This paper proposes Noisy Kernel, a new framework to model the label noise, which generalizes the bounded noise setting studied by prior work. The main result of this paper is to show that the Hellinger gap of the noisy kernel gives a good characterization of the min-max mistake bound of learning a finite hypothesis class under their model. Specifically, they design a learner with mistake bound $\log^2(|H|)/\gamma_H$, where $\gamma_H$ is the Hellinger gap of the noise kernel. On the other hand, for any fixed noisy kernel, they design a hypothesis class $H$ such that every learner has a mistake bound at least $\log(|H|)/\gamma_H$ to learn $H$. For the binary classification problem, the author further designs a learner with a mistake bound $\log(|H|)/\gamma_L$, where $\gamma_L$ is the $L^2$ gap of the noisy kernel, improving the dependence on $\log(|H|)$.

**Strengths:**

1. This paper proposes a novel framework that models the label noise for the online classification problem. This generalizes the previous model for bounded label noise.
2. To prove the upper bound for the mistake bound, the paper introduces a novel reduction from the classification problem to the pairwise hypothesis testing problem, which could potentially be useful for designing algorithms for other learning problems.
3. A matching lower bound on the dependence on the Hellinger gap of the noisy kernel is proved.

**Weaknesses:**

1. Though the framework proposed by the paper models a large class of noise models, the learner designed by the paper only works for learning finite hypothesis classes.
Prior work, Agnostic Online Learning by Ben-David et al. proves a mistake bound that depends on the Littlestone dimension of the hypothesis class. There is a big gap between hypothesis classes with finite Littlestone dimension and finite hypothesis classes. For example, the class of singleton functions is infinite but has Littlestone dimension 1. Such a simple class is not learnable using the learner designed by the paper because it is impossible to do pairwise hypothesis testing for an unbounded set of hypotheses.
From this point of view, I do not think the paper fully answers which class is learnable under their noisy model.
Perhaps, this might be because the noise model studied by the paper is too general and is thus hard for one to design algorithms that make use of the structure of the hypothesis class.

2. Though the paper gives a tight bound on the dependence on the Hellinger gap of the noise kernel, the upper bound and the lower bound differ by a $\log(|H|)$ factor, which is sometimes linear in the dimension of the input and thus cannot be ignored in general. Though the paper shows that for binary classification, they can close this gap, a new gap on the dependence of the divergence gap is created.

**Questions:**

1. The model of the noisy kernel is not as intuitive as the prior noisy models such as Massart noise. In section 5, the authors mention that the noisy kernel model has many applications such as learning with privacy constraints or learning from quantum data. What would be the noisy kernels for these types of problems?

2. To run the learner designed by the paper, one has to know the error bound of the pairwise hypothesis testing problem. How can one know such an error bound before running the online learning algorithm? If we do not know such a bound, would it be possible to get any data-driven algorithm?

3. I would like the authors to comment on the weakness pointed out above.

---

> ### Author Rebuttal · Authors · 2024-07-31
>
> We thank the reviewer for the detailed review and helpful comments. We now address the main concerns raised below:
>
> **Infinite Classes:** In fact, as mentioned in Remark 1, our techniques also work for *infinite classes*. Indeed, for any class $\mathcal{H}$ of finite Littlestone dimension $\mathsf{Ldim}(\mathcal{H})$, we can apply our algorithm on the *sequential* experts as in Section 3.1 of [1] (specifically the multi-class version from Daniely et al., 2015, Thm. 25) to arrive at an upper bound $O(\frac{\mathsf{Ldim}^2(\mathcal{H})\log^2 (NT)}{\gamma_H})$. This follows because our pairwise testing framework works for the *sequential experts* as well, which is *finite* and of size $(NT)^{\mathsf{Ldim}(\mathcal{H})+1}$. By construction of the experts, any risk achieved against the (finite) sequential experts implies the same risk for the (infinite) class $\mathcal{H}$. See also our general rebuttal and responses to reviewers crkZ and JscE for further consequences.
>
> **Tightness on $\log |\mathcal{H}|$:** Indeed, our logarithmic factor can result in an additional factor for Littlestone dimension. This is introduced by the aggregating of pairwise testings in Algorithm 1. We would like to emphasize that pairwise testing is an essential technique for dealing with *multi-class* label classes. We are unaware of any prior technique that can even lead to *sublinear risk* for the multi-class cases. Therefore, we view our result as significant even with a suboptimal logarithmic factor, as it provides the first known systematic treatment of this problem.
>
> **Questions:**
>
> 1. Consider the general template in our Example 2, which can be viewed as a "generalized" Massart noise in the multiclass case. For the multi-class *randomized response mechanism* in differential privacy, please refer to our global rebuttal for a concrete construction. To realize the quantum setting, one may consider the task of classifying quantum states into certain classes (i.e., our true labels). Here, the kernel set for any given label would be the collection of all distributions induced by certain measurement on the quantum states corresponding to that class.
>
> 2. Indeed, Algorithm 1 needs to know the upper bound of the pairwise testing error $C$. This error bound is completely determined by the Hellinger gap of the kernel and not the realization of the data; please refer to our proof of Theorem 2. Since the Hellinger gap is an intrinsic complexity measure of the problem, it can be used as an input for the algorithm. We believe that investigating a data-dependent way of selecting the parameter $C$ is also of significant interest. In fact, using a multiplicative weighted majority algorithm with a substantially more complicated potential-based analysis, we can show that an $O(C^2+C\log(CK))$ risk bound holds without the knowledge of $C$. However, it is unclear to us whether this bound is tight for unknown parameter $C$.
>
> [DSBS15]  A. Daniely, S. Sabato, S. Ben-David, and S. Shalev-Shwartz. Multiclass Learnability and the ERM Principle. JMLR 2015.

---

> > ### Comment · Reviewer_GopC · 2024-08-09
> >
> > Thanks for the response. I raised my score to 6.

---

### Official Review · Reviewer_crkZ · 2024-07-07

**Soundness:** 3
**Presentation:** 3
**Contribution:** 3
**Rating:** 6
**Confidence:** 3

**Summary:**

This paper studies online classification where the features are chosen adversarially, but the labels are corrupted by some unknown stochastic noise. The learner makes predictions using the true features and noisy labels, but is evaluated against the true labels (noiseless) in the minimax sense. The authors consider a very general noise generation mechanism specified by a kernel, which for every (feature, true label) pair, outputs a subset of distributions over the noisy label space. The adversary generates the noisy label by picking a distribution output by the kernel and sampling from it. The authors study the minimax risk under this setting and derive upper and lower bounds on the minimax risk in terms of the Hellinger gap of the noisy distributions output by the kernel and the cardinality of the hypothesis class. The upper and lower bounds differ only by a log factor in the cardinality of the hypothesis class, showing that (up to log factors), the Hellinger gap is the correct complexity measure of the kernel that characterizes the minimax rates. The authors prove these results by developing a novel reduction from noise online classification to hypothesis testing. Along the way, they provide guarantees for a new conditional version of Le Cam-Birge testing in the online setting.

**Strengths:**

- The paper is well-written and easy to follow
- The problem of noisy online classification is well-motivated (but also see weakness below)
- The technical contributions are interesting and the proof techniques are novel. In particular, I found the reduction to pairwise hypothesis testing to be nice
- Overall, I think this paper makes a solid contribution to the literature on online classification

**Weaknesses:**

- More clarification is needed about the problem setup beyond just stating the minimax value. From what I can get, the adversary is oblivious and must pick the entire sequence of features, true labeling hypothesis, and noisy labels before the game begins. However, there seems to be some subtlety in the choice of how the adversary can pick the noisy labels which I feel requires explanation in the main text. In particular, the adversary can select noisy label distributions based on the realized values of the previous nosy labels. This contrasts from a weaker adversary which has to pick the entire sequence of noisy label distributions up front, and then samples the entire sequence of noisy labels.

- Lack of motivation for generalization. While it is nice to provide results for a generalized noise mechanism, ideally there is some sort of justification for the generalization beyond theoretical curiosity (i.e why should I care). It would be nice if you can provide concrete example of real-world situations (i.e. specifying the instance space, label, space, noisy label space, kernel etc.) which are captured by your generalization, but not by previous noise models. As far as I can tell, applications to privacy, online denoting, and physical measurements are mentioned, but not expanded upon. Overall, I think it would be nice to provide more motivation behind why I should care about generalizing the noise mechanism.

- Lack of universal lower bounds. While the upper bounds are stated for every (finite) hypothesis class, the lower bounds are of the form "there exists a hypothesis class such that the minimax value is at least...". It would nice to have universal lower bounds of the form "for every hypothesis class, the minimax value is at least...".

- Finite Hypothesis Class. This paper currently only studies the minimax rates for finite hypothesis classes. There is a remark indicating that the results extend to infinite hypothesis classes via covering techniques from [1, 21], however neither the proof of this statement, nor the resulting minimax rates are provided. I am not super convinced that these existing covering techniques can give you bounds on the minimax value that are independent of the sizes of the true and noisy label spaces (like your existing bounds are). For example, for the simply hypothesis class $H = \{ x \mapsto a: a \in [N] \}$, to cover a single point $x \in X$, you need the entire hypothesis class $H$.  It would be nice if you can give the upper and lower bounds on the minimax value for infinite hypothesis classes that your results imply along with a proof sketch.

Please see questions below regarding each of these weakness.

**Questions:**

- Is my interpretation of the minimax value correct? If so, I think it would be nice to make this explicit in text.
- Can you give me some concrete learning problems for which your results provide new insights (i.e can you justify the generalization of the noise mechanism)?
- Can you comment about universal lower bounds in your setting?
- Can you comment on the generalization of your results to infinite hypothesis classes? Is it true that using your results with existing covering arguments would result in factors of N and M (the cardinalities of true and noisy label spaces) appearing in your bounds?
- Right now, your bounds are written in terms of two separate complexity measures - one on the hypothesis class (i.e its cardinality) and one on the kernel (i.e. well-separated). Could there be a single complexity measure, a function of both the hypothesis class and kernel, that better captures the minimax value? If so, do you have any intuition on what this might be or what difficulties arise when trying to do so?
- Your existing results assume that the kernel outputs sets of distributions that are convex and closed. It would be nice if you can make explicit where exactly this assumption is being used in the main text. In addition, what can be said if you are guaranteed that the kernel always outputs a finite set of distributions (say of size at most G)? Can you comment on how would the minimax rates would scale with G?

**Limitations:**

No limitations.

---

> ### Author Rebuttal · Authors · 2024-07-31
>
> We thank the reviewer for the detailed review and helpful comments. We now address the main concerns raised below:
>
> **Adaptivity w.r.t. Noisy Labels:** Indeed, in our setup, the noisy label distributions are selected *adaptively*, while the features are selected obliviously w.r.t. the realization of noisy labels. In fact, the features can also be made adaptive to noisy labels for the risk bound in Theorem 5. However, oblivious features are required in our *conditional* Le Cam Birgé testing, i.e., Theorem 4 (although we believe this can be resolved using a more tedious minimax analysis that involves both $x^J$ and $y^J$). Therefore, for the clarity of presentation, we have assumed that the features are generated obliviously w.r.t. noisy labels. We will make this clearer.
>
> Please note that the oblivious assumption for noisy label distributions and features w.r.t. the *learner's prediction* is not essential by a standard argument (c.f. Appendix H), provided the predictions have *independent* internal randomness among different time steps (which is satisfied by our algorithms).
>
> **Motivation for Generalization:** One of the primary motivations for generalization is to deal with cases with multi-class labels, where prior techniques based on EWA algorithms do not apply. A concrete example is that of Example 2. Note that we can instantiate Example 2 to the multi-class *randomized response mechanism* in differential privacy by specifying the $p_y$ as the singleton distribution that assigns probability 1 on $y\in \mathcal{Y}$ (with $\mathcal{Y}=\tilde{\mathcal{Y}}$)  and the TV-radius is $\frac{N}{e^{\epsilon}-1+N}$ (to achieve $(\epsilon,0)$ local differential privacy).  To our knowledge, the risk bound implied by our result is novel for this particular differential private online setting. Please refer to our global rebuttal for more details.
>
> **Universal lower bounds:** Indeed, our lower bound only holds in the minimax sense. Although, an $\Omega(\frac{1}{\gamma_H})$ lower bound holds for *any* non-trivial classes (i.e., there exist at least two distinct functions).
>
> **Infinite Classes:** Indeed, for any class $\mathcal{H}$ of finite Littlestone dimension $\mathsf{Ldim}(\mathcal{H})$, we can apply our algorithm on the experts constructed in Section 3.1 of [1] (specifically the multi-class version) to arrive at an upper bound $O(\frac{\mathsf{Ldim}^2(\mathcal{H})\log^2 (NT)}{\gamma_H})$ (independent of the noisy label set size $M$). Note that a logarithmic dependency on $N$ is necessary for finite Littlestone dimension classes (in the worst case). To see this, consider the class $H$ constructed by the reviewer. It is easy to see that the class has Littlestone dimension 1. However, one can assign the noisy label distributions to each $a\in \mathcal{Y}$ with constant pairwise KL-divergences (using, e.g., distributions of the form $p[m]=\frac{1\pm \epsilon}{M}$ for $m\in \tilde{\mathcal{Y}}$ with $M \sim\log N$). The $\Omega(\log N)$ lower bound will then follow by Fano's inequality. This is in contrast to the noiseless setting where the regret is independent of label set size.
>
> **Unified measure:** As mentioned above, we conjecture that the correct growth might be $O(\frac{\mathsf{Ldim}(\mathcal{H})\log(NT)}{\gamma_H})$ for any "non-trivial" class $\mathcal{H}$ and kernels independent of features. However, a general unified measure seems to be far from being achievable using our current techniques. This is due to the complicated structure of $\mathcal{H}$ and the weakness of Le Cam's two-point method (used in our lower bound proof). We believe investigating such a unified complexity measure would be of significant interest for future research.
>
> **Convexity Assumption:** The convexity assumption is used in the Le Cam Birgé testing (for the minimax theorem to work). Please note that the convexity is *without loss of generality* in our setting, since the adversary is allowed to choose the distribution arbitrarily, including a randomized strategy that is effectively selected from the convex hull. Therefore, any distributional strategy that the adversary might choose can be represented within this convex framework.

---

> > ### Comment · Reviewer_crkZ · 2024-08-07
> >
> > Thanks for your response.

---

### Official Review · Reviewer_JscE · 2024-07-12

**Soundness:** 3
**Presentation:** 4
**Contribution:** 2
**Rating:** 6
**Confidence:** 3

**Summary:**

The paper generalized an agnostic online learning setting from [1]: Nature sequentially produces arbitrary feature vectors, and a noiseless label chosen from an arbitrary hypothesis within a given class. The learner observes the feature vector (exactly), and a noisy label chosen from a distribution chosen from a class of distributions, where the identity of the class depends on the noiseless label. The goal of the learner is to minimize the expected cumulative risk. The paper's main result (Theorem 2, also presented in a simplified form in Theorem 1) is characterization of the high probability (upper bound) and expected risk (lower bound) in terms of the Hellinger gap. The result follows from a proposed learning algorithm, which is based on pairwise testing. In addition, assuming a positive $L_2$ gap, tighter upper bounds are obtained w.r.t. the cardinality size of the hypothesis class (Theorem 5).

**Strengths:**

1. Theorem 2 captures the exact dependence on the Hellinger distance $\gamma_H$, by providing upper and lower bounds whose dependence on that parameter is $\Theta(1/\gamma_H)$.

2. An additional result based on $L_2$ distance, which is tight also in the log-cardinality of the hypothesis class.

3. The introduction of the surrogate pairwise loss function and evaluating the loss by its maximum over all competing hypothesis, as well as the resulting reduction of online learning to binary hypothesis testing is interesting.

**Weaknesses:**

My main issue with the paper is that given the separation condition and finite hypothesis class, the resulting learning task is “easy”:

1. The assumption of a strictly positive Hellinger distance significantly eases the learning task. The immediate consequence of this is that the cumulative risk is bounded in $T$, and also hints the ease of the learner's task. As the algorithm also suggests, the problem is reduced to a composite binary hypothesis testing using $T$ samples, followed by static play of that decision.

2. Given that the problem is interesting, it is worth to already address the infinite class setting, and noises that are not uniformly bounded. While the technical details *might* be routine, the dependency of the risk bounds should be interesting, and, in relation to the previous comment, might also lead to elaborated risk bounds, with interesting dependency on the function class (parametric/a-parametric) and the noise conditions.

**Questions:**

1. What is the conclusion from Example 1? From the paragraph that precedes it, it appears that it meant to justify either the semi-stochastic setting or the evaluation of the risk on the true labels, but it is not discussed how. From reading the rest of the paper, it appears that the analysis in this paper is a generalization of the result in Example 1, but this is not an intrinsic motivation.

2. Proof of Theorem 2: To the best of my knowledge, such problem is called *composite* hypothesis testing (rather than robust), and this was extensively studied in the information-theoretic literature. In this regard, the error in hypothesis testing is related to the KL divergence, for which chain rule is much more suitable to distributions that are not in product form via the standard chain rule. My question is why can't the analysis be made in terms of the KL divergence ?

Finally, regarding line 41, what is differential piracy ? Only attacking some ships ? :-)

**Limitations:**

The paper properly discusses limitations throughout.

---

> ### Author Rebuttal · Authors · 2024-07-31
>
> We thank the reviewer for the detailed review and helpful comments. We now address the main concerns raised below:
>
> **Non-triviality of problem setup:** We would like to clarify that the non-triviality of our pairwise testing scheme lies in how to *use* the testing results in an *online* fashion, as the testing results are not reliable (only the tests involving the ground truth are reliable, but it is *unknown* a priori). Moreover, the pairwise testing is not actually "static" since the composite distribution sets depend on the features, which are unknown as well. In fact, even for the Bernoulli noise case, the proof (c.f. Ben-David et al., Theorem 15 [1]) relies on quite non-trivial *backward induction*. It is not clear at all from their proof how the denominator $1-2\sqrt{\eta(1-\eta)}$ arises. One of our primary contributions is to figure out the *precise* complexity measure (i.e., the Hellinger gap) that determines the learning complexity and to provide a clearer characterization of the underlying paradigm.
>
> We would like to emphasize that the Hellinger gap (and the noisy kernel formulation) is not actually an "assumption" but a *consequence* of our characterization. It is our kernel formulation that allows us to approach the noisy online classification problem in such a clean manner, considering that prior proof techniques (c.f. [1]) are quite obscure even for Bernoulli noise.
>
> **Non-uniform gap and infinite classes:** In fact, as  mentioned in Remark 1, our technique also works for *infinite classes* and *non-uniform* gaps. Indeed, for any class $\mathcal{H}$ of finite Littlestone dimension $\mathsf{Ldim}(\mathcal{H})$, we can apply our algorithm on the *sequential* experts constructed in Section 3.1 of [1] (specifically the multi-class version) to arrive at an upper bound $O(\frac{\mathsf{Ldim}^2(\mathcal{H})\log^2 (NT)}{\gamma_H})$. For binary-valued classes, Theorem 5 also implies an $O(\frac{\mathsf{Ldim}(\mathcal{H})\log T}{\gamma_L})$ upper bound. Moreover, for the *non-uniform* gaps, such as Tsybakov noises of order $\alpha$ as in [8], our results (primarily Proposition 1 and Theorem 3) implie a risk of order $\tilde{O}(T^{\frac{2(1-\alpha)}{2-\alpha}})$, and this is tight up to poly-logarithmic factors.
>
> We have chosen to omit these results in the submission due to limited space and their incremental nature, but we are happy to include them in the final version (given the additional page).
>
> **Questions:**
>
> 1. Example 1 is meant to provide a concrete instance of how stochastically generated labels (i.e., with Massart noise) can lead to non-trivial risk bounds on true labels that are independent of the time horizon, even though the noise rate is linear in T. Therefore, it is natural to investigate to what extent such a phenomenon can occur in more general noisy settings and to find the precise complexity measure that characterizes learnability. See also our global rebuttal for examples on differential privacy.
>
> 2. We thank the reviewer for clarifying the terminology. Our Hellinger-based testing error bound is essentially a *conditional* version of the Theorem 32.8 from [17]. We are unaware of any (general) analysis based on KL-divergences (we would greatly appreciate it if the reviewer could point it out to us). Since the KL-divergence is *not* symmetric, it is not well-suited to our case since we need to measure the *distance* between *sets*.
>
> **Typos:** We thank the reviewer for catching the typo; the correct term should be "differential privacy."

---

> > ### Comment · Reviewer_JscE · 2024-08-14
> >
> > Thank you for the clarifications.
> > I understand that the Hellinger is a result of the analysis, but it is a consequence of the fact that the problem is just reduced to testing. It is not very surprising that testing results that does not involve the ground truth do not prevail the true hypothesis.
> > So, overall I am still not fully convinced that the problem goes well beyond hypothesis testing, but it is nonetheless an interesting theoretical contribution, and so I have raised my score.

---

### Author Rebuttal · Authors · 2024-07-31

We thank all of the reviewers for their helpful comments. We would like to clarify the following two concerns shared by most of the reviewers:

**Infinite Classes:** As mentioned in Remark 1, our techniques also work for *infinite classes*. Indeed, for any class $\mathcal{H}$ of finite Littlestone dimension $\mathsf{Ldim}(\mathcal{H})$, we can apply our algorithm on the *sequential experts* as constructed in Section 3.1 of [1] (specifically the multi-class version from Daniely et al., 2015, Thm 5) to arrive at an upper bound $O(\frac{\mathsf{Ldim}^2(\mathcal{H})\log^2 (NT)}{\gamma_H})$. This is because, for any finite Littlestone dimensional class, we can always find a *finite* sequential cover of size $(NT)^{\mathsf{Ldim}(\mathcal{H})+1}$; therefore, it can be effectively reduced to the finite classes case resolved in our work.

**Real-World Applications:** We would like to clarify that our work is not intended to introduce any specific noise models. Rather, our main focus is to provide a *characterization* to determine, for any given noise model, the *best achievable* minimax risk. Since our algorithmic approach is general, it will serve as a baseline for various application scenarios.

To give a clear exposition on how this works, we consider the following *online local differential privacy* setting. Let $\mathcal{H}$ be a hypothesis class with label set $\mathcal{Y}$ of size $N$. The *randomized response mechanism* works as follows: for any $y \in \mathcal{Y}$, we generate $\tilde{y}$ being $y$ with probability $1-\eta$ and generate *uniformly* from $\mathcal{Y}$ with probability $\eta$, where $\eta \le \frac{N}{e^{\epsilon}+N-1}$ (which achieves $(\epsilon, 0)$-local differential privacy). In this case, the noisy label set equals the true label set and the *noisy kernel* maps each $(x, y)$ to the distribution set $\\{(1-\eta)e_y + \eta u : \eta \leq \frac{N}{e^{\epsilon} + N - 1}\\}$ where $e_y$ assigns probability 1 to $y$ and $u$ is uniform over $\mathcal{Y}$. It is not hard to verify that the Hellinger gap of the kernel is $\Theta(\frac{\epsilon^2}{N})$ (for sufficiently small $\epsilon$). Our Theorem 2 then implies the risk bound $O(\frac{N\log^2|\mathcal{H}|}{\epsilon^2})$. To our knowledge, the implied risk bound is original for this particular differentially private multi-class online setting.

[DSBS15]  A. Daniely, S. Sabato, S. Ben-David, and S. Shalev-Shwartz. Multiclass Learnability and the ERM Principle. JMLR 2015.

---

### Decision · Program_Chairs · 2024-09-25

**Decision:**

Accept (poster)

**Comment:**

A borderline paper. It is a theory paper on online classification with a very general stochastic label noise corruption and the input features can be generated adversarially. The paper characterizes learning rates by a notion of Hellinger gap and reduction to a testing problem. The reviews agree that the paper is well-written and the contribution interesting, especially the reduction to pairwise testing and the fact that the model can be seen as a generalization of previous noise models (eg Massart noise). On the other hand, the results for now only apply to the case of a finite hypothesis class, and while the authors discussed convincingly that their results could be extended to infinite classes (using Littlestone dimension), they recognize that the complexity of the class (or log-cardinality in the case of a finite class) only appears in the upper bound and not the lower bound, resulting in a significant theoretical gap. The request for experimental validation by one reviewer seems out of place given that the paper's focus is precise theory.

Overall weak accept recommendation.